# PTADisc: A Cross-Course Dataset Supporting Personalized Learning in Cold-Start Scenarios

**Liya Hu**[1], **Zhiang Dong**[1], **Jingyuan Chen**[1,2*], **Guifeng Wang**[1], **Zhihua Wang**[2], **Zhou Zhao**[1] and **Fei Wu**[1,2*]

[1]Zhejiang University
[2]Shanghai Institute for Advanced Study of Zhejiang University
{liyahu, dongza, jingyuanchen, wangguifeng, zhihua.wang, zhaozhou,
wufei}@zju.edu.cn

## Abstract

The focus of our work is on diagnostic tasks in personalized learning, such as cognitive diagnosis and knowledge tracing. The goal of these tasks is to assess students' latent proficiency on knowledge concepts through analyzing their historical learning records. However, existing research has been limited to single-course scenarios; cross-course studies have not been explored due to a lack of dataset. We address this issue by constructing **PTADisc** , a **D**iverse, **I**mmense, **S**tudent-centered dataset that emphasizes its sufficient **C**ross-course information for personalized learning. PTADisc includes 74 courses, $1,530,100$ students, $4,054$ concepts, $225,615$ problems, and over $680$ million student response logs. Based on PTADisc, we developed a model-agnostic **C**ross-**C**ourse **L**earner **M**odeling **F**ramework (**CCLMF**) which utilizes relationships between students' proficiency across courses to alleviate the difficulty of diagnosing student knowledge state in cold-start scenarios. CCLMF uses a meta network to generate personalized mapping functions between courses. The experimental results on PTADisc verify the effectiveness of CCLMF with an average improvement of 4.2% on AUC. We also report the performance of baseline models for cognitive diagnosis and knowledge tracing over PTADisc, demonstrating that our dataset supports a wide scope of research in personalized learning. Additionally, PTADisc contains valuable programming logs and student-group information that are worth exploring in the future.

## 1 Introduction

Personalized learning aims to provide students with customized learning services that align with their specific goals and abilities, which is facilitated by the vast amount of data accumulated on thriving online learning platforms. The focus of personalized learning is on diagnosing students' knowledge state and involves two key research tasks: (1) Cognitive Diagnosis (CD) [23, 5, 24, 3, 27, 8, 13], which assesses students' static latent proficiency on concepts using their learning records, and (2) Knowledge Tracing (KT) [21, 32, 20, 9, 30, 29, 16, 25, 15], which evaluates students' dynamic latent proficiency during various study phases based on their past sequential learning records. Based on the diagnostic results of students' knowledge states, several applications such as Computerized Adaptive Testing [1, 14] and Personalized Educational Planning [11] can be conducted, as illustrated in Figure 1(a).

To support the above research, several educational datasets are constructed [7, 12, 4, 31, 33, 22, 28]. However, existing datasets still face the following challenges: (1) Insufficient Data Coverage: Many

---

* Corresponding author.

37th Conference on Neural Information Processing Systems (NeurIPS 2023) Track on Datasets and Benchmarks.

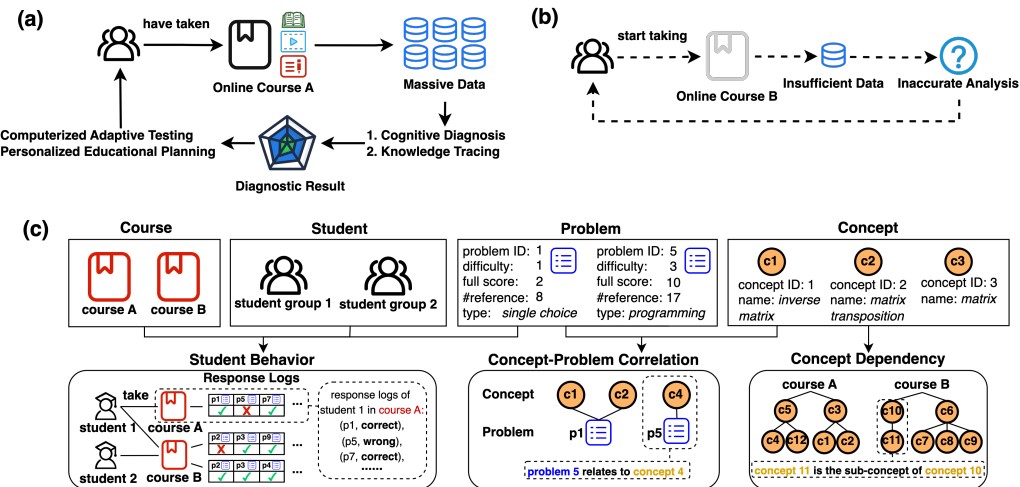

Figure 1: (a) Illustration of the personalized learning process. (b) Illustration of cold-start problem in personalized learning. (c) Overview of PTADisc.

existing datasets are limited in their application scope due to being built for specific tasks. For instance, KDD Cup 2010 [12] lacks exercise content text, making it unsuitable for supporting content-based knowledge tracing. (2) Lack of Concept Annotation: Existing datasets reveal two problems including incomplete relation annotations between concepts and problems, and missing dependency relations among concepts, as seen in Table 5. (3) Poor Cross-course Information: Most existing datasets are constructed within a single course (*e.g.*, ASSIST [7] in *Math* and EdNet [4] in *English*). Although MOOCCubeX [31] consists of 4,216 courses, there exist no students with learning records in multiple courses. Cross-course studies are not supported by existing datasets.

In our work, we first address the above challenges by constructing a Diverse, Immense, Student-centered and Cross-course dataset licensed by **P**rogramming **T**eaching **A**ssistant (**PTA**[1]) platform, namely **PTADisc**. Specifically, PTADisc features in: (1) **D**iverse. As shown in Figure 1(c), it contains various information, comprising four key entities and relationships among them. The entities include course, student, problem, and concept. The relationship among entities is reflected in student behavior, concept-problem correlation and concept dependency. (2) **I**mmense. It includes 74 courses, 1,530,100 students, 4,504 concepts, 225,615 problems, and over 680 million student exercising response logs (*i.e.*, answer correctly or not). (3) **S**tudent-centered. The entire dataset is organized around student behaviors (*i.e.*, response logs), providing valuable insight into personalized learning. (4) **C**ross-course. A subdataset with 29,454 students simultaneously taking 5 courses is extracted. This makes PTADisc **the first dataset to support cross-course analysis**.

Furthermore, we proposed a model-agnostic **C**ross-**C**ourse **L**earner **M**odeling **F**ramework (**CCLMF**) based on the cross-course subdataset of PTADisc. As shown in Figure 1(b), when a student enrolls in a new course and has few response logs, it's difficult to predict the student's latent proficiency on concepts and future performance. To address this cold-start problem, CCLMF leverages student latent proficiency relationships between courses to transfer knowledge from courses with sufficient response logs, thereby improving performance in the target course. The experimental results demonstrate the advancement of CCLMF in cold-start scenarios with an average improvement of 4.2% on AUC. Our code and datasets are available at `https://github.com/wahr0411/PTADisc.git`.

The contributions of this paper include: (1) Construct PTADisc, a diverse, immense, student-centered and cross-course dataset, supporting various studies in personalized learning. (2) Construct a cross-course subdataset with a significant amount of students enrolled in multiple courses, making up for the lack of existing datasets that cannot support cross-course analysis. (3) Propose CCLMF, a model-agnostic Cross-Course Learner Modeling Framework which can improve the performance of diagnostic tasks in cold-start scenarios.

---

[1] `https://pintia.cn/`

## 2 Problem Definition

This study focuses on two tasks: cognitive diagnosis (CD) and knowledge tracing (KT). CD aims to analyze students' latent proficiency on concepts, assuming that they are in a stable learning state. Alternatively, KT focuses on dynamically assessing the knowledge proficiency of students as they progress through the learning process. Figure 2 visually represents the differences.

### 2.1 Cognitive Diagnosis

Suppose there is a course with a total of $N$ students, $M$ problems, and $K$ knowledge concepts, which can be denoted as $\mathcal{S} = \{s_1, s_2, \ldots, s_N\}$, $\mathcal{P} = \{p_1, p_2, \ldots, p_M\}$ and $\mathcal{C} = \{c_1, c_2, \ldots, c_K\}$ respectively. The response logs of student $s$ are denoted by $\mathcal{R}_s$, a set of tuple $(p, r)$, with $p \in \mathcal{P}$ and $r$ indicating the score obtained by student $s$ on problem $p$. In addition, we have $\boldsymbol{Q} = \{Q_{ij}\}_{M \times K}$, with $Q_{ij} = 1$ if

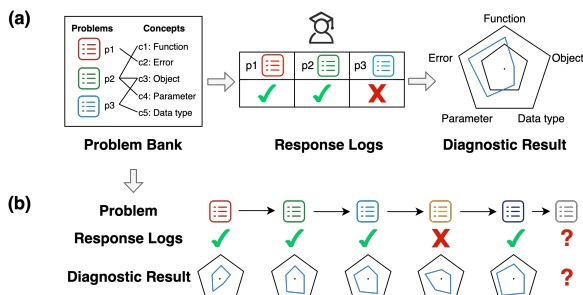

Figure 2: Illustration of CD (a) and KT (b).

problem $p_i$ is related to knowledge concept $c_j$, and $Q_{ij} = 0$ in all other cases. The goal of CD is to assess student's level of proficiency on different knowledge concepts through the prediction process of student performance, given students' response logs $\mathcal{R}$ and the Q-matrix $\boldsymbol{Q}$ [27].

### 2.2 Knowledge Tracing

The presentations of problems and knowledge concepts in KT are the same as in CD. For each student $s$, their response logs are represented by $\mathcal{R}_s = \{\boldsymbol{e}_j\}_{j=1}^l$, where $\boldsymbol{e}_j$ denotes the $j$-th response log and $l$ is the total number of logs. Each $\boldsymbol{e}$ is denoted by a triplet $(p, r, t)$ where $p \in \mathcal{P}$, $r$ is the score and $t$ is the timestamp of the student's response. The goal of KT is to predict students' proficiency levels of concepts at different study phases, given their response logs $\mathcal{R}$ and the Q-matrix $\boldsymbol{Q}$.

## 3 Dataset

PTADisc is sourced from PTA, an online learning platform developed by Hangzhou PAT Education Technology Co., Ltd. PTA is an automatic program evaluation and open teaching assistance platform for universities and society. Given the close collaboration between PTA and universities, it is common for students to concurrently pursue a series of courses that align with their training program. Up to July 2023, PTA has attracted over $1,000$ organizations, $9,000$ teachers and $3,900,000$ users and provides a problem bank of over $290,000$ problems referenced by course problem sets and exams. The highlight of PTA is that it covers a significant amount of students enrolled in multiple courses. This feature perfectly meets the need to conduct cross-course research and mine student characteristics between courses.

### 3.1 Privacy Protection

To prevent privacy disclosure, we have excluded personal and sensitive data such as student names and email addresses, retaining only the unique student IDs as individual identifiers, with anonymization employed. Additionally, we confirmed that the user-generated data was strictly authorized during the registration process, as specified in the *terms of service* and *privacy statement* of the PTA platform.

### 3.2 Dataset Construction

#### 3.2.1 Raw Data Processing

First, we manually selected 74 courses. Then as shown in Figure 3, we applied privacy protection, data curation, data filtering

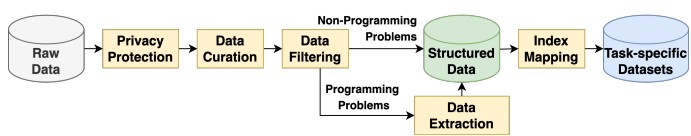

Figure 3: Construction pipeline of PTADisc.

and data extraction to generate intermediate structured data. To eliminate redundancy, we created a unified problem bank that captured each problem's full score, content, and relevant concepts across all courses. Additionally, we filtered out problems without concept annotations or those with full scores of 0. For non-programming data, we selected only those students who had 5 or more response logs for each course. For programming data, we performed extraction to include relevant details such as codes submitted, programming language, as well as time and memory consumption.

### 3.2.2 Structured Data

The data is organized by course, and we provide examples of course, problem, and response logs in Tables 1, 2 and 3, respectively. These structured data include the following content.

**Knowledge Concepts.** Knowledge concepts, also known as knowledge skills, are structured into a tree format that illustrates the hierarchical dependencies among concepts, as depicted in Figure 1(c). This hierarchical structure is manually annotated using the textbook catalog, and each relation is denoted as a tuples $(c_i, c_j)$ where $c_j$ is the sub-concept of $c_i$. Each problem in the dataset is associated with one or more leaf concepts within the tree, while the course name is the root node.

**Problems.** On PTA platform, registered teachers can create and publish problems containing content, title, problem difficulty, problem type, full score, and other specialized configurations. In general, PTADisc contains two broad categories of problems, which are non-programming and programming problems. All the problems are stored in a problem bank and assigned a unique problem ID.

**Student Groups.** Student groups can be regarded as classes. Each student group is given a student group ID and contains multiple students taking the same course.

**Problem Sets.** Problem sets are published by teachers as homework or quiz. All the problems in problem sets are selected from the problem bank. Once a problem is selected into a problem set, it is given a problem_set_problem ID (psp ID) and a specified full score. Each problem set has an opening and closing time, and students are only allowed to complete the problem set during this period.

**Student Behaviors.** Response logs are provided to represent student behaviors, including submission time, problem type, test score, psp ID, and judge status. It is important to note that PTADisc provides a variety of judge-related information for the programming problems, such as code, language, running time and memory consumption, which are well worth exploring in the future.

Table 1: An example of a course in PTADisc.

| Course ID | Course Name | Concept ID | Concept Name | Concept Parent ID | Problem ID | Student Group ID | Student ID |
|---|---|---|---|---|---|---|---|
| C_9088 | Python Programming | C_1568 | Loop | C_9088 | P_3122 | G_2144 | S_e28d |
| | | C_9472 | Function | C_9088 | P_3120 | G_2144 | S_369e |
| | | C_2592 | Break | C_1568 | P_2600 | G_5952 | S_1c3a |
| | | C_7488 | Continue | C_1568 | P_3143 | G_5952 | S_6f58 |

Table 2: Examples of two problems in PTADisc.

| Problem ID | Concept ID | Difficulty | Reference Count | Problem Type | Problem Set Problem ID | Full Score | Problem Set ID | Start Time | End Time |
|---|---|---|---|---|---|---|---|---|---|
| P_9600 | C_3408 | 3 | 1479 | Programming | PSP_1617 | 10 | PS_8480 | 2018/6/7 01:08 | 2018/7/6 23:59 |
| | | | | | PSP_3168 | 15 | PS_4512 | 2018/7/3 11:50 | 2018/7/8 23:59 |
| P_4480 | C_5696, C_6400 | 1 | 464 | Multiple choice | PSP_7537 | 2 | PS_2112 | 2020/11/3 08:24 | 2020/11/4 10:24 |
| | | | | | PSP_5952 | 1 | PS_2448 | 2021/3/26 16:00 | 2021/3/31 23:59 |

Table 3: Examples of three response logs in PTADisc.

| Problem Type | Submission ID | Student ID | Submit Time | Score | Problem Set Problem ID | Status | Language | Code | Judge Logs |
|---|---|---|---|---|---|---|---|---|---|
| True or false | Sub_4736 | S_9059 | 2018/12/28 09:05 | 0 | PSP_4731 | NO_ANSWER | \ | \ | \ |
| Multiple choice | Sub_1520 | S_9059 | 2018/12/28 08:58 | 2 | PSP_4750 | ACCEPTED | \ | \ | \ |
| Programming | Sub_5088 | S_be7f | 2018/6/8 19:18 | 75 | PSP_6944 | PARTIAL_ ACCEPTED | PYTHON | # Code s=input() ...... | time: 0.025, memeory: 3260416, result: ACCEPTED; ...... |

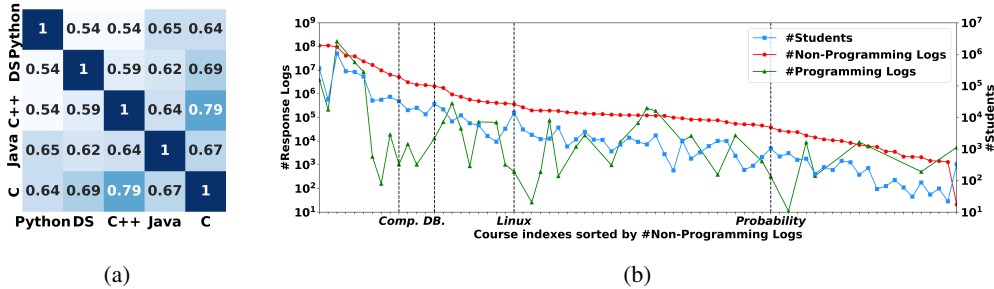

| (a) | (b) |

Figure 4: PTADisc statistics. (a) Correlation coefficients of student performance between five courses. (b) The distribution of response logs and enrolled students.

### 3.2.3 Task-specific Datasets

**Cognitive Diagnosis Dataset.** Following the generation of structured data, we performed index mapping to create the CD datasets, which consist of a Q-matrix and students' response logs. The Q-matrix stores the relationships between problems and concepts. The response logs of each student were split into training, validation, and testing datasets in a ratio of 70%/10%/20%, respectively.

Table 4: Cross-course datasets in PTADisc.

| Course Name | Java | C++ | Python | DS[2] | C |
|---|---|---|---|---|---|
| #Students | 29,454 | 29,454 | 29,454 | 29,454 | 29,454 |
| #Problems | 16,752 | 15,172 | 17,787 | 21,952 | 26,056 |
| #Response Logs | 4,750,970 | 6,587,356 | 6,454,336 | 7,789,280 | 11,378,017 |
| #Concepts | 773 | 547 | 685 | 767 | 847 |

**Knowledge Tracing Dataset.** The process for generating the KT dataset is similar to the CD dataset, except that each log in the KT dataset includes an additional submitting time.

**Cross-course Dataset.** We analyzed students taking multiple courses and identified the top five frequent course sets. Furthermore, we eliminated students with less than 10 response logs for any of the five courses and were left with a total of 29,454 students for the study, as presented in Table 4.

### 3.3 Dataset Statistics

PTADisc includes 74 courses, 1,530,100 students, 4,504 concepts, 225,615 problems, and over 680 million student response logs. The response logs were partitioned according to problem type into programming and non-programming logs. Figure 4(b) presents the distribution of response logs and enrolled students for each course. Additional detailed statistics can be found in Appendix B.

### 3.4 Dataset Characteristics

The comparison of PTADisc with other open-access educational datasets is shown in Table 5. We divide these datasets into three categories: (1) student-centered datasets including ASSIST09[7][3], Junyi[4], KDD Cup 2010 [12], EdNet [4], which focus on student behaviors and are frequently used for diagnostic tasks in personalized learning; (2) knowledge-centered datasets including MOOC-CubeX [31]; and (3) programming datasets including BePKT [33] and CodeNet [22]. We illustrate the characteristics of PTADisc in four aspects.

**Diverse.** As illustrated in Table 5, PTADisc contains extensive concept-related information and detailed records of student behaviors. PTADisc offers coverage of concept dependencies which are not found in ASSIST, EdNet and CodeNet. Moreover, PTADisc is the only dataset that evaluates student responses using a scoring ratio system rather than binary values [17]. Besides, PTADisc includes student group information, enabling group-level analysis.

**Immense.** As Table 5 illustrates, PTADisc is currently the largest educational dataset, featuring diverse data scales across multiple courses. This range of data scales presents researchers with a multitude of options for personalized learning studies.

---

[2]DS stands for *Data Structure and Algorithm Analysis*.

[3]https://sites.google.com/site/assistmentsdata/datasets

[4]https://pslcdatashop.web.cmu.edu/DatasetInfo?datasetId=1198

Table 5: Comparison between PTADisc and existing open datasets.

| Field | Dataset Name | PTADisc | ASSIST | Junyi | KDD. | EdNet | MOOCCubeX | BePKT | CodeNet |
|---|---|---|---|---|---|---|---|---|---|
| Overview | Number of Logs | **6.81e8** | 4.02e5 | 2.59e7 | 2.27e6 | 1.31e8 | 2.96e8 | 6.79e4 | 1.39e7 |
| | **Cross-course Info** | ✓ | × | × | × | × | × | × | × |
| Concepts Variety | Concepts Annotations | ✓ | ✓ | ✓ | ✓ | ✓ | incomplete[5] | ✓ | × |
| | Concepts Dependencies | ✓ | × | ✓ | × | × | ✓ | × | × |
| Problems Variety | Programming Problems | ✓ | × | × | × | × | × | ✓ | ✓ |
| | Problem Type | ✓ | ✓ | ✓ | × | × | ✓ | × | × |
| Logs Variety | Detailed Programming Logs | ✓ | × | × | × | × | × | ✓ | ✓ |
| | Non-binary Response Results | ✓ | × | × | × | × | × | × | × |
| Other Info | Problem Set | ✓ | ✓ | × | ✓ | ✓ | ✓ | × | × |
| | Student Group | ✓ | ✓ | ✓ | × | × | × | × | × |

**Student-centered.** PTADisc is a student-centered dataset focusing on student behaviors. By validating problems and knowledge concepts through analysis of student response logs, the dataset improves consistency and maintainability, making it well-suited for diagnostic tasks.

**Cross-course.** PTADisc is the first dataset with cross-course information. For the cross-course subset in PTADisc, we further analyzed the correlation coefficient between these courses based on the performance of each student. As shown in Figure 4 (a), there is a positive correlation among the five courses, providing a statistical basis for cross-course learner modeling.

### 3.5 Dataset Applications

PTADisc provides support for CD and KT tasks, as mentioned in Section 2. Section 5 presents the CD and KT baseline experimental results on several existing methods. Additionally, PTADisc can also support the following tasks in the field of AI for education: (1) Prerequisite discovery [19], which identifies the sequence in which concepts or topics should be learned, ensuring foundational concepts are understood before advanced ones. (2) Computerized adaptive testing [1], which aims to rapidly and accurately diagnose a candidate's level of knowledge mastery through personalized test items. (3) Educational recommendation [11], which provides appropriate learning suggestions to students based on their interactions with problems. (4) Cross-course research. PTADisc provides cross-course information, enabling the study of how students perform in different classes. We conducted a cross-course study addressing the cold-start issue in Section 4.

With various information provided, PTADisc also has the potential to support the following research directions: (1) It provides non-binary performance data. Most existing CD and KT methods tackle this as binary classification (wrong/right answer). Non-binary grades allow for regression-focused investigations. (2) It provides information on problem types, aiding research of CD and KT. (3) It provides information on problem difficulty, enabling to study how the level of difficulty of exercises relates to students' final learning outcomes. (4) It provides problem set specifics, like submission time, enabling modeling of students' learning habits based on when they submit their work. (5) It provides data on student groups, facilitating the assessment of teaching quality within classes and group-level educational analysis. (6) It provides information on programming exercises, including detailed code submissions and records, enabling in-depth research into programming-related studies.

## 4  CCLMF: Cross-course Learner Modeling Framework

By utilizing the cross-course subdataset of PTADisc, we can leverage the relationships between courses to mitigate the challenges associated with the cold-start problem. When a student begins a new course (*i.e.*, target course) and has limited data, it can be challenging to diagnose his proficiency level through CD or KT. To address this problem, we present the **C**ross-**C**ourse **L**earner **M**odeling **F**ramework (**CCLMF**) inspired by cross-domain recommendation [18, 34], which utilizes auxiliary information from another course (*i.e.*, source course) taken by the student and has enough data. By identifying connections between the student's proficiency in the target and source courses, CCLMF enhances the model's performance in low-data scenarios. CCLMF incorporates a meta-learner to predict network parameters, leveraging the power of meta-learning to improve performance in the target course by harnessing the knowledge acquired from source courses. Please note that CCLMF

---

[5]Concept annotations are not available for all problems, since the annotation process is similar to keyword matching.

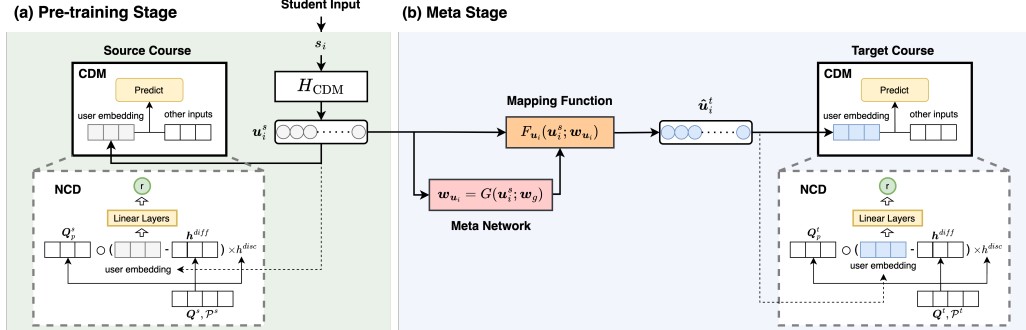

Figure 5: CCLMF architecture consists of two stages. (a) In the **pre-training stage**, we train a CDM in the source course and get student $i$'s proficiency representation $\boldsymbol{u}_i^s$. (b) In the **meta stage**, we utilize the meta network to generate the mapping function $F_{\boldsymbol{u}_i}$ for each student $i$, and the proficiency representation in the target course is obtained by $\boldsymbol{u}_i^t = F_{\boldsymbol{u}_i}(\boldsymbol{u}_i^s; \boldsymbol{w}_{\boldsymbol{u}_i})$.

---

**Algorithm 1** CCLMF: Cross-course Learner Modeling Framework

---

**Input:** shared students: $\mathcal{S}$
    other inputs: $\mathcal{P}^s, \mathcal{C}^s, \boldsymbol{Q}^s, \mathcal{R}^s, \mathcal{P}^t, \mathcal{C}^t, \boldsymbol{Q}^t, \mathcal{R}^t$
**Pre-training Stage:**
    1. Train a CDM in the source course.
    2. Get student proficiency representation in the source course: $\boldsymbol{u}_i^s = H_{\mathrm{CDM}}(s_i; \boldsymbol{w}_h)$.
**Meta Stage:**
    1. Use a meta network $G(\cdot)$ to generate mapping function $F_u$.
    2. Get the student proficiency representation in the target course by $\hat{\boldsymbol{u}}_i^t = F_{\boldsymbol{u}_i}(\boldsymbol{u}_i^s; \boldsymbol{w}_{\boldsymbol{u}_i})$.
    3. Train $G(\cdot)$ and $F_u$ by optimizing $loss_{\mathrm{CC-CDM}} = loss_{\mathrm{CDM}}(\hat{\boldsymbol{u}}_i^t, other\_inputs)$.
**Inference:**
    For a cold-start student $s_j$ in the target course, his level of proficiency is $F_{\boldsymbol{u}_j}(\boldsymbol{u}_j^s; \boldsymbol{w}_{\boldsymbol{u}_j})$.

---

is a model-agnostic framework that can be applied to various CD or KT models. To explain it in a simpler way, we will illustrate CCLMF in the context of cognitive diagnosis.

## 4.1 Problem Definition

We assumed that $N$ students have enrolled in both the source and target courses, denoted as $\mathcal{S}$. The source course has $M^s$ problems and $K^s$ knowledge concepts, denoted as $\mathcal{P}^s$ and $\mathcal{C}^s$. And the target course has $M^t$ problems and $K^t$ knowledge concepts, denoted as $\mathcal{P}^t$ and $\mathcal{C}^t$. The response logs in the source and target course are denoted as $\mathcal{R}^s$ and $\mathcal{R}^t$. For both courses, the Q-matrices are denoted as $\boldsymbol{Q}^s = \{Q_{ij}^s\}_{M^s \times K^s}$ and $\boldsymbol{Q}^t = \{Q_{ij}^t\}_{M^t \times K^t}$. The goal of the CCLMF is to accurately measure students' level of proficiency on various knowledge concepts in the target course, which incorporates student response logs and Q-matrices from both the target and source courses.

## 4.2 CCLMF Architecture

We illustrate the architecture of CCLMF in Figure 5. CCLMF mainly consists of two stages: a pre-training stage and a meta stage. In the pre-training stage, a cognitive diagnosis model (CDM) such as NCD is trained using data from the source course, yielding a representation $\boldsymbol{u}_i^s$ for student $s_i$. In the meta stage, instead of directly mapping $\boldsymbol{u}_i^s$ to student representation of the target course $\boldsymbol{u}_i^t$, we applied the idea of meta-learning to learn a personalized cross-course transformation function $F_{\boldsymbol{u}}$ for each student. Moreover, this meta learning procedure is task-oriented, with $F_{\boldsymbol{u}}$ being trained by minimizing Equation (4) for specific tasks using data in the target course. After completing the meta stage, a personalized transformation function is obtained for each student. Thus, given any student $s_j$ who is new to the target course, their proficiency representation can be determined by their transformation function as $F_{\boldsymbol{u}_j}(\boldsymbol{u}_j^s; \boldsymbol{w}_{\boldsymbol{u}_j})$.

Table 6: CCLMF results on MIRT and NCD.

| Metrics | Model | no dropout | 10% dropout | 20% dropout | 30% dropout | 40% dropout | 50% dropout |
|---|---|---|---|---|---|---|---|
| AUC | MIRT | 0.6379 | 0.6412 | 0.6398 | 0.6399 | 0.6342 | 0.6363 |
| | CC-MIRT | 0.7059 (+0.0680) | 0.7025 (+0.0612) | 0.6998 (+0.0600) | 0.6998 (+0.0599) | 0.6957 (+0.0615) | 0.6918 (+0.0555) |
| ACC | MIRT | 0.6832 | 0.7037 | 0.6869 | 0.6886 | 0.6889 | 0.6991 |
| | CC-MIRT | **0.7854 (+0.1022)** | 0.7834 (+0.0797) | 0.7826 (+0.0958) | 0.7833 (+0.0947) | 0.7822 (+0.0933) | 0.7797 (+0.0806) |
| RMSE | MIRT | 0.4902 | 0.4821 | 0.4884 | 0.4916 | 0.5013 | 0.4949 |
| | CC-MIRT | 0.3948 (-0.0954) | 0.3965 (-0.0856) | 0.3973 (-0.0911) | 0.3973 (-0.0943) | 0.3983 (-0.1029) | 0.3999 (-0.0950) |
| AUC | NCD | 0.6885 | 0.6846 | 0.6797 | 0.6787 | 0.6736 | 0.6675 |
| | CC-NCD | 0.7106 (+0.0221) | 0.7061 (+0.0215) | 0.7028 (+0.0231) | 0.7007 (+0.0221) | 0.6953 (+0.0216) | 0.6895 (+0.0221) |
| ACC | NCD | 0.7613 | 0.7662 | 0.7682 | 0.7606 | 0.7635 | 0.7640 |
| | CC-NCD | 0.7859 (+0.0246) | 0.7834 (+0.0172) | 0.7819 (+0.0137) | 0.7814 (+0.0208) | 0.7812 (+0.0177) | 0.7817 (+0.0177) |
| RMSE | NCD | 0.4109 | 0.4095 | 0.4081 | 0.4127 | 0.4121 | 0.4133 |
| | CC-NCD | 0.3966 (-0.0143) | 0.3973 (-0.0122) | 0.3981 (-0.0100) | 0.3991 (-0.0136) | 0.3998 (-0.0123) | 0.4009 (-0.0124) |

**Pre-training stage.** CCLMF aims to leverage valuable insights gleaned from the vast data in the source course. In the pre-training stage, a CDM is trained in the source course, generating an informative student representation $\boldsymbol{u}^s$. Specifically, our CCLMF can be employed on CDMs which characterize student proficiency as:

$$\boldsymbol{u}_i^s = H_{\mathrm{CDM}}(s_i; \boldsymbol{w}_h), \tag{1}$$

where $\boldsymbol{w}_h$ denotes the parameters of $H_{\mathrm{CDM}}$, and $H_{\mathrm{CDM}}$ is a function abstracted from CDM which can calculate each student's latent proficiency representation $\boldsymbol{u}_i^s$. The dimensionality of vector $\boldsymbol{u}_i^s$ is $d^s$ which is usually related to the number of knowledge concepts and the number of problems. The generated $\boldsymbol{u}^s$ contains personalized and auxiliary information that can be utilized in the meta stage.

**Meta stage.** Due to individual differences, the relationships between student proficiency in the source and target courses can vary significantly from one student to another. Therefore we need to create personalized mapping functions for each student in order to retain students' individual characteristics. In the meta stage, we utilized a meta network to learn personalized mapping functions for each student. The meta network $G(\cdot)$ is formulated as:

$$\boldsymbol{w}_{\boldsymbol{u}_i} = G(\boldsymbol{u}_i^s; \boldsymbol{w}_g), \tag{2}$$

where $\boldsymbol{w}_g$ is the parameters of $G(\cdot)$ and $\boldsymbol{w}_{\boldsymbol{u}_i}$ is used as the parameters of the mapping function. The personalized mapping function $F_{\boldsymbol{u}}(\cdot; \boldsymbol{w}_{\boldsymbol{u}})$ then produces personalized transformed student's representation in the target course as:

$$\boldsymbol{u}_i^t = F_{\boldsymbol{u}_i}(\boldsymbol{u}_i^s; \boldsymbol{w}_{\boldsymbol{u}_i}). \tag{3}$$

Instead of mapping-oriented optimization used by Man et al.[18], we directly used the performance of diagnostic tasks as our optimization goal to train the meta network. This task-oriented training procedure advances in making full use of the ground truth values rather than approximate intermediate results. Therefore, the meta network $G(\cdot)$ as well as the mapping function $F_{\boldsymbol{u}}(\cdot; \boldsymbol{w}_{\boldsymbol{u}})$ are trained together using data in the target course.

Given the ground truth value $r$ from $\mathcal{R}^t$ and the CC-CDM's final output $\hat{r}_i$ which is generated based on $\boldsymbol{u}_i^t$, the task-oriented loss can be formulated as:

$$loss_{\mathrm{CC-CDM}} = -\sum_i (r_i \log \hat{r}_i + (1 - r_i) \log (1 - \hat{r}_i)). \tag{4}$$

**Inference.** The goal of CCLMF is to accurately measure students' level of proficiency in the target course. During inference, for any student $s_j$ who is new to the target course, their level of proficiency can be determined by their personalized transformation function $\boldsymbol{u}_j^t = F_{\boldsymbol{u}_j}(\boldsymbol{u}_j^s; \boldsymbol{w}_{\boldsymbol{u}_j})$.

The whole procedure of CCLMF is summarized in Algorithm 1. Detailed implementation of CCLMF on NCD [27] can be found in Appendix C.

### 4.3 Experiment Settings

We conducted CCLMF on a traditional cognitive diagnosis model MIRT [24] and a deep-learning-based model NCD [27], called CC-MIRT and CC-NCD respectively.

**Datasets.** We constructed the cross-course datasets based on the datasets shown in Table 4. As depicted in Figure 4(a), the correlation coefficient between students' performance in *Python Programming* (Python) and *Java Programming* (Java) is $0.65$, indicating a relatively high correlation. This high correlation makes these two courses well-suited for addressing cold-start problems. *Python Programming* was chosen as the source course. To simulate cold-start scenarios, $332,568$ response logs ($7\%$ of each student's response logs) from *Java Programming* were selected to form the target course. We reserved $20\%$ of response logs in the target course as the test set and then conducted experiments on sub-datasets with varying degrees of sparsity. Specifically, we randomly dropped $10\%$, $20\%$, $30\%$, $40\%$, and $50\%$ of the remaining data and split the resulting data into 7/1 as train/valid set. To reduce the influence of randomness, we repeated the dropout process 10 times for each dropout ratio, then reported the average results across the 10 sets of data.

**Settings and Metrics.** The meta network was implemented as a two-layer perceptron with input-size $K^s$, output-size $K^s \times K^t$ and hidden-size 100, where each linear layer was followed by a RELU activation function. The weights were initialized by Xavier [10]. The batch size and learning rate are 128 and $1 \times 10^{-3}$. All experiments were executed on a Linux server with two GeForce RTX 3090s. The evaluation metrics include Area Under the ROC Curve (AUC) [2], Prediction Accuracy (ACC) and Root Mean Square Error (RMSE).

## 4.4 Experimental Results and Analysis

Table 6 shows the superiority of CCLMF over baseline models. CCLMF has better performance on both NCD and MIRT models with different dropout ratios on all metrics. Specifically, CCLMF achieves an average improvement of $4.2\%$ on AUC, $5.5\%$ on Accuracy, and $5.3\%$ on RMSE.

To investigate the performance of our model over students of different sparsity levels, we show the performance with respect to the number of response logs a student has in Figure 6. Note that we did not re-train the model with different sets of students, instead we divided the test set into different groups by the number of logs per student. We observe that the performance improvement of CCLMF is more significant for students with fewer response logs, highlighting the advantage of our model in cold-start scenarios.

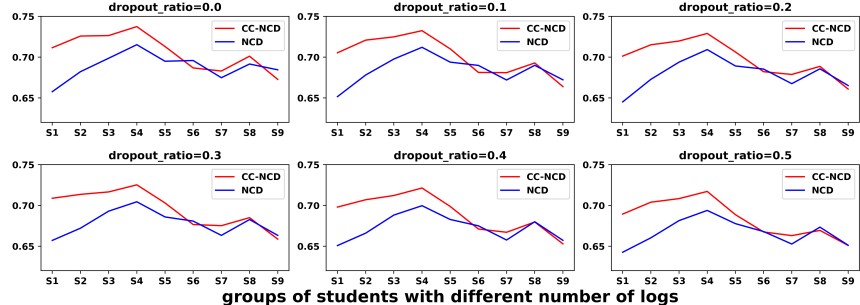

Figure 6: AUC on NCD and CC-NCD w.r.t. student groups with varying degrees of sparsity.

## 5 Experiments on CD and KT Tasks

To demonstrate PTADisc's support for CD and KT, we specifically selected four courses with varying response log scales, namely *Probability and Statistics* (Probability), *Linux System* (Linux), *Database Technology and Application* (DB), and *Computational Thinking* (Comp), as seen in Table 7, which were also marked in Figure 4(b) with vertical dashes.

Table 7: Datasets for CD and KT.

| Course Name | Probability | Linux | DB | Comp |
|---|---|---|---|---|
| #Students | 557 | 4,398 | 12,646 | 45,329 |
| #Problems | 1,054 | 2,678 | 3,616 | 8,399 |
| #Concepts | 247 | 284 | 325 | 477 |
| #Response Logs | 46,106 | 391,434 | 2,363,206 | 6,504,414 |
| Logs per Student | 82.78 | 89.00 | 186.87 | 143.49 |

### 5.1 Baseline Models

For CD, we considered the following baseline methods: three traditional methods **DINA** [5, 26], **IRT** [6], and **MIRT** [24], deep-learning method **NCD** [27] and graph based method **RCD** [8]. For

KT, we considered the following baseline methods: deep sequential method **DKT [21]**, memory augmented method **DKVMN [32]**, attention-based methods **AKT [9]** and **SAKT [20]**, graph based methods **GIKT [30]** and **SGKT [29]**, and pre-training based method **PEBG [16]**. Due to the intensive computing resources required by RCD, we did not conduct RCD on two courses with relatively large-scale data, which are DB and Comp. Additional information of these baseline models can be found in Appendix D.

## 5.2 Experiment Settings and Metrics

For each baseline model, we employed the same parameter settings and optimization methods as described in their respective papers to ensure fairness in the comparison. The evaluation metrics used for CD included AUC, ACC, and RMSE, while for KT, the evaluation metrics were AUC and ACC, which are consistent with commonly used metrics in the literature.

## 5.3 Experimental Results and Analysis

The results of the CD and KT tasks have yielded notable findings, as shown in Table 8. Firstly, we observed that the dataset size has a significant influence on model performance, with higher prediction accuracy observed for larger datasets. Secondly, we noted in Table 8 that the MIRT model performance was generally higher than that of the NCD model; this contrasts with the findings in Table 6. We attribute this discrepancy to the cross-course dataset's lower average number of student response logs (*i.e.* approximately 10 logs per student), compared to the larger dataset with over 80 logs per student, as shown in Table 7. These results suggest that the number of logs per student significantly influences model performance, and that the NCD model may outperform the traditional MIRT model specifically in cases where there are fewer student logs on average. Thirdly, PEBG achieved the best AUC performance across all datasets in the KT task, which demonstrates the effectiveness of employing pre-training techniques to capture the heterogeneous educational data.

Table 8: Baselines of existing CD and KT models on PTADisc.

| Dataset Name | Metric | DINA | IRT | MIRT | NCD | RCD | DKT | DKVMN | SAKT | AKT | GIKT | SGKT | PEBG |
|---|---|---|---|---|---|---|---|---|---|---|---|---|---|
| Probability | AUC | 0.6569 | 0.7257 | 0.7324 | 0.7092 | **0.7485** | 0.6876 | 0.6815 | 0.6879 | 0.7123 | 0.7086 | 0.7079 | **0.7320** |
|  | ACC | 0.6121 | 0.7097 | 0.7112 | 0.6894 | **0.7196** | **0.7025** | 0.6961 | 0.6721 | 0.6978 | 0.6883 | 0.6938 | 0.6977 |
|  | RMSE | 0.5141 | 0.4490 | 0.4730 | 0.4590 | **0.4346** | - | - | - | - | - | - | - |
| Linux | AUC | 0.7577 | 0.8199 | 0.8168 | 0.8171 | **0.8318** | 0.7898 | 0.7856 | 0.7756 | 0.8074 | 0.8173 | 0.8156 | **0.8379** |
|  | ACC | 0.7053 | 0.7802 | 0.7799 | 0.7755 | **0.7860** | 0.7726 | 0.7713 | 0.7665 | 0.7849 | 0.7791 | 0.7801 | **0.8040** |
|  | RMSE | 0.4493 | 0.3900 | 0.4023 | 0.3934 | **0.3841** | - | - | - | - | - | - | - |
| DB | AUC | 0.7141 | 0.7901 | **0.8121** | 0.7901 | - | 0.7813 | 0.7635 | 0.7562 | 0.7956 | 0.8101 | 0.7996 | **0.8381** |
|  | ACC | 0.7856 | 0.8322 | **0.8424** | 0.8299 | - | 0.8312 | 0.8281 | 0.8181 | **0.8383** | 0.8353 | 0.8299 | 0.8373 |
|  | RMSE | 0.3934 | 0.3493 | **0.3407** | 0.3504 | - | - | - | - | - | - | - | - |
| Comp | AUC | 0.7137 | 0.7819 | **0.8096** | 0.7734 | - | 0.7978 | 0.7811 | 0.7717 | 0.8091 | 0.8091 | 0.8172 | **0.8281** |
|  | ACC | 0.7303 | 0.7929 | **0.8018** | 0.7880 | - | 0.8276 | 0.8234 | 0.8012 | 0.8283 | **0.8337** | 0.8274 | 0.8194 |
|  | RMSE | 0.4335 | 0.3808 | **0.3749** | 0.3849 | - | - | - | - | - | - | - | - |

## 6 Conclusion

**Outlook**: We describe concrete ongoing and future work based on PTADisc. Specifically,

- We will conduct research about adaptive learning and personalized educational planning, and incorporate them into a personalized learning system alongside the CCLMF model as shown in Figure 1(a). We will also analyze the group-level student learning behaviors.

- We will explore programming knowledge tracing based on PTADisc which contains a large amount of multi-round programming problem submission records and rich evaluation information.

**Limitation**: The original content text of the problems cannot be shared currently due to copyright constraints. We intend to disclose the content by extracting text features in the future.

**Conclusion**: PTADisc is a diverse, immense, student-centered and cross-course dataset that enables researchers to conduct previously infeasible cross-course studies. Based on PTADisc, we developed CCLMF to alleviate the difficulty of diagnosing student knowledge states in the cold-start scenario. Furthermore, we demonstrate the broad range of applications of our dataset by reporting on the performance of baseline models for CD and KT tasks over PTADisc.

## Acknowledgments and Disclosure of Funding

The authors wish to acknowledge the generous support and contributions of Hangzhou PAT Education Technology Co. Ltd. for this research. This work was supported by the National Natural Science Foundation of China (No.62037001, No.62307032, No.62293550), Shanghai Rising-Star Program (23QA1409000), National Key R&D Program of China (No. 2022ZD0117104), and the Starry Night Science Fund at Shanghai Institute for Advanced Study (SN-ZJU-SIAS-0010).

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

# PTADisc: A Cross-Course Dataset Supporting Personalized Learning in Cold-Start Scenarios (Appendix)

## Contents

Our code, dataset, and detailed description for the dataset are available at `https://github.com/wahr0411/PTADisc.git`.

## A  Related Work

### A.1  Cognitive Diagnosis and Knowledge Tracing

The goal of cognitive diagnosis (CD) is to assess students' level of proficiency of different knowledge concepts through the prediction process of student performance, given students' exercise records, *aka* response logs. Traditional psychometric-based methods include Item Response Theory (IRT) [18], MIRT [19] and DINA [4, 22]. These methods depend on manually designed functions and the effectiveness requires a large number of psychological experiments to verify, which are labor-intensive and lack of ability to capture complex relationships between students, problems, and knowledge concepts. Recent years, with the development of artificial intelligence [10], deep learning-based cognitive diagnosis models have been developed [23, 6, 9, 28]. Specifically, NCD [23] incorporates neural networks to learn the complex exercising interactions. And RCD [6] models the interactive and structural relations via a multi-layer student-problem-concept relation map.

The goal of knowledge tracing (KT) is to dynamically model students' knowledge proficiency through her historical learning records, so as to predict her performance to new problems. Traditional probabilistic KT models assume students' knowledge state as a set of binary variables where each variable represents whether a student masters an individual concept or not, such as Bayesian Knowledge Tracing (BKT) [3]. Recent years, deep learning-based KT models are proposed for learning valid representations especially when large amounts of data are available, such as DKT [21] and DKVMN [26]. Further, from the perspective of model structure, a few methods based on transformers (SAKT [15]), GNNs (SGKT [24]), and pre-training frameworks (PEBG [11]) are proposed.

### A.2  Cross-Domain Recommendation

Cross-domain recommendation is a promising method to alleviate data sparsity and the cold-start problem [30, 27, 2]. Several models have been proposed, including CMF [20], which uses shared parameters for all domains, and CST [14], which transfers knowledge about users and items from auxiliary data sources. Mapping-based methods have been shown to be effective in solving cold-start recommendation problems [12], by learning a mapping function from the source domain to the target domain. However, these methods have limited generalization ability for cold-start items or users. To address this issue, TMCDR [29] introduces meta learning to improve the generalization ability and PTUPCDR [30] further improves TMCDR by learning personalized bridges for each user. While the cross-domain problem has been widely explored in the recommendation domain, there is limited research on cross-course learner modeling in personalized learning.

### A.3  Other educational applications

**Prerequisite discovery** refers to the task of identifying and establishing the sequence or order in which concepts or topics should be learned or presented, ensuring that foundational concepts are understood before more advanced ones [13]. Suppose a MOOC corpus is composed by $n$ courses in the same subject area, denoted as $\mathcal{D} = \{\mathcal{D}_1, \cdots, \mathcal{D}_i, \cdots, \mathcal{D}_n\}$, where $\mathcal{D}_i$ signifies an individual course. Course concepts are subjects taught in the course, i.e., the concepts not only mentioned but also discussed and taught in the course. Let us denote the course concept set of $\mathcal{D}$ as $\mathcal{C} = \mathcal{C}_1 \cup \cdots \cup \mathcal{C}_n$, where $\mathcal{C}_i$ representing the concepts intrinsic to $\mathcal{D}_i$. Prerequisite relation learning in MOOCs is formally defined as follows. Given a MOOC corpus $\mathcal{D}$ and its corresponding course concepts $\mathcal{C}$, the objective is to learn a function $\mathcal{P} : \mathcal{C}^2 \to \{0, 1\}$ that maps a concept pair $\langle a, b \rangle$, where $a, b \in \mathcal{C}$, to a binary class that predicts whether $a$ serves as a foundational prerequisite for concept $b$.

**Computerized adaptive testing (CAT)** is an emerging testing format in many standardized examinations, aiming to rapidly and accurately diagnose a candidate's level of knowledge mastery through personalized test items [1]. Let's conceptualize a set of students represented by $\mathcal{S} = \{s_1, s_2, \ldots, s_N\}$, a problem set represented by $\mathcal{P} = \{p_1, p_2, \ldots, p_M\}$ and a set of knowledge concepts represented by $\mathcal{C} = \{c_1, c_2, \ldots, c_K\}$ related to the problems. We denote the record of student $s_i$ answering problem $p_j$ as a triplet $r_{ij} = \langle s_i, p_j, a_{ij} \rangle$, where $a_{ij}$ equals 1 if $s_i$ answers $p_j$ correctly, and 0 otherwise. Problem set $\mathcal{P}$ is divided into a tested set $\mathcal{P}_T$ and an untested set $\mathcal{P}_U$. When introduced to a novel student $s_i \in \mathcal{S}$, a problem pool $\mathcal{P}$ with knowledge concepts $\mathcal{C}$, the challenge is to architect a strategy $\mathcal{A}$ to select a $X$-size question set $\mathcal{P}_T = \{p_1^*, p_2^*, \ldots, p_X^*\}$ step by step that has the maximum quality and diversity. Prior to the testing phase, we set up an abstract cognitive diagnosis model $\mathcal{M}$ with

parameters $\theta$ capturing knowledge states. During testing, at step $t(1 \leq t \leq X)$, we select one question $p_t^* = \mathcal{A}(\mathcal{P}_U, \mathcal{M})$, then observe a new interaction test record $r_{it}^* = \langle s_i, p_t^*, a_{it}^* \rangle$ and update the knowledge states, i.e., $\boldsymbol{\theta}$, in $\mathcal{M}$ instantly. After testing, we measure the effectiveness of $\mathcal{A}$ by computing $\mathrm{Inf}(\mathcal{A})$ and $\mathrm{Cov}(\mathcal{A})$, where $\mathrm{Inf}(\mathcal{A})$ denotes the measurement of quality and $\mathrm{Cov}(\mathcal{A})$ denotes the measurement of diversity.

**Educational recommendation** lies in constructing a recommend system that can process the interactions between students and questions. This system should be capable of making appropriate learning suggestions to students [8]. In the context of a digital educational platform, assume there are $N$ students and $P$ problems. We record the exercising process of a certain student $n = \{(p_1, r_1), (p_2, r_2), \cdots, (p_t, r_t)\}, n \in \mathcal{N}$, where $p_t \in \mathcal{P}$ represents the problems that student $n$ practices at her time step $t$, and $r_t$ denotes the corresponding score. Conventionally, a correct response to problem $p_t$ is denoted by $r_t$ equals to 1, , and an incorrect response by $r_t$ equals to 0. Each problem $p \in \mathcal{P}$ is characterized by a triplet $p = \{w, c, d\}$. Specifically, the element $w$ represents its text content as a word sequence $p = \{w_1, w_2, \ldots, w_W\}$. $c \in \mathcal{C}$ describes its knowledge concept coming from all $K$ concepts. And $d$ means its difficulty factor.

## B   Dataset Statictics

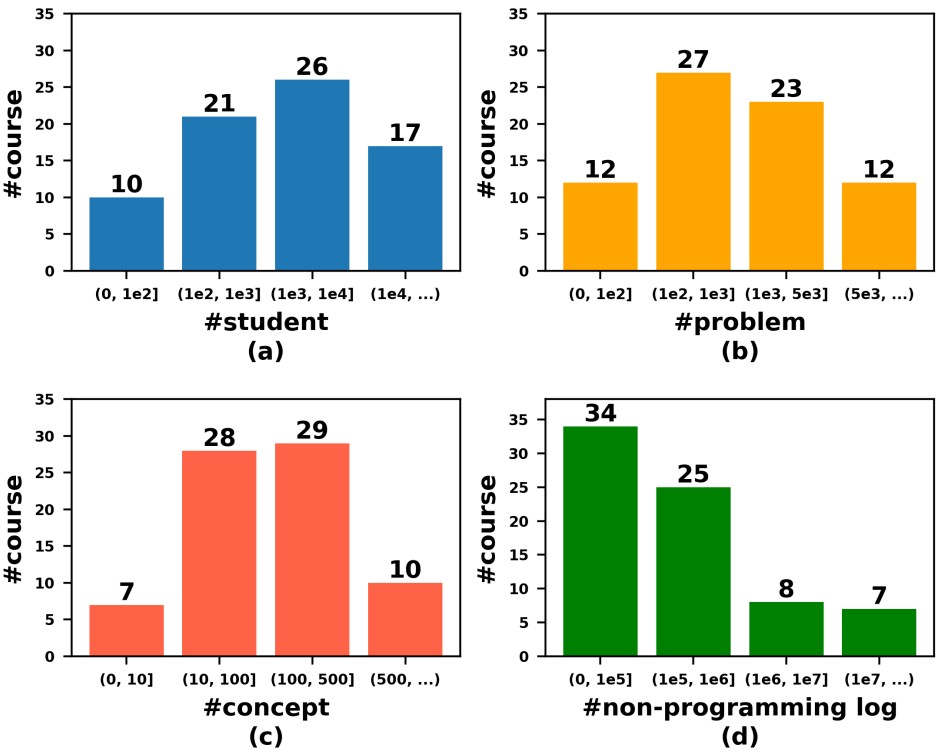

Figure 1: Distribution of the number of students, problems, concepts and non-programming logs over courses in PTADisc.

Table 1: Detailed statistics of 68 courses in PTADisc.

| course name | #student | #problem | #concept | #non-programming log | #programming log |
|---|---|---|---|---|---|
| C++ programming | 362,585 | 20,786 | 617 | 110,641,195 | 4,058,080 |
| Computer App. foundation | 37,280 | 7,304 | 527 | 110,200,887 | 216,146 |
| C programming | 1,074,901 | 43,140 | 1,069 | 97,385,606 | 164,459,382 |
| DS. and algorithm analysis | 294,236 | 29,914 | 897 | 41,667,618 | - |
| Python programming | 277,621 | 28,396 | 817 | 39,087,014 | 21,827,297 |
| Java programming | 198,335 | 24,524 | 906 | 23,542,208 | 8,461,930 |
| Computational thinking foundation | 34,519 | 4,500 | 401 | 16,871,472 | 2,212 |
| Information technology | 36,700 | 6,499 | 516 | 9,935,894 | 159 |
| Computational thinking | 45,329 | 8,399 | 477 | 6,504,414 | 19,085 |
| Database principle | 33,031 | 10,428 | 758 | 5,219,955 | 1,039 |
| Information processing technology and App. | 17,152 | 2,428 | 221 | 3,127,821 | 7,536 |
| Computer network | 20,333 | 7,372 | 609 | 2,451,121 | 1,023 |
| Database technology and App. | 12,646 | 3,616 | 325 | 2,363,206 | - |
| Introduction to computer science | 26,884 | 4,077 | 419 | 2,108,904 | 13,159 |
| Operating system | 18,237 | 8,563 | 726 | 1,777,233 | 64,947 |
| Object oriented programming java | 7,552 | 2,993 | 315 | 944,474 | 395,830 |
| Principles of computer composition | 11,699 | 5,474 | 475 | 755,982 | 34,456 |
| Web front-end technology | 6,612 | 4,247 | 438 | 572,879 | 876 |
| Compilation principle | 5,635 | 1,830 | 263 | 493,449 | 65,249 |
| Thinking by data | 2,597 | 1,027 | 156 | 445,636 | - |
| Multivariate statistical analysis | 1,704 | 790 | 83 | 416,427 | 62,393 |
| Linux system | 4,398 | 2,672 | 284 | 391,434 | 1,026 |
| Computer and problem solving | 14,241 | 1,518 | 226 | 370,605 | 522 |
| Software engineering | 4,197 | 2,815 | 335 | 270,692 | - |
| Assembly language programming | 2,830 | 1,300 | 206 | 200,231 | 26 |
| Machine learning | 2,073 | 1,984 | 331 | 197,685 | 500 |
| Csharp programming | 2,134 | 1,981 | 91 | 194,984 | 76,053 |
| Java web | 4,783 | 1,547 | 215 | 189,171 | 332 |
| Big data processing technology | 1,211 | 1,055 | 155 | 165,573 | - |
| VB programming | 2,032 | 1,273 | 113 | 157,360 | 5,829 |
| Discrete mathematics and App. | 3,516 | 1,278 | 123 | 147,522 | 17,855 |
| Digital image processing | 1,984 | 956 | 218 | 142,497 | - |
| English | 1,952 | 660 | 19 | 139,328 | - |
| Software project management | 850 | 1,753 | 133 | 133,824 | 970 |
| Scala programming | 1,378 | 616 | 124 | 127,382 | 9,367 |
| Literature and history | 2,306 | 886 | 31 | 126,748 | - |
| Discrete mathematics | 1,694 | 608 | 92 | 125,378 | 58,724 |
| Fortran programming | 1,398 | 858 | 106 | 123,498 | 248,698 |
| Intro to algorithm competition | 2,699 | 1,403 | 229 | 119,565 | 190,772 |
| Data warehouse and data mining | 689 | 777 | 97 | 117,717 | - |
| Practice of statistics | 210 | 394 | 35 | 99,217 | - |
| Principles of information security | 1,844 | 1,298 | 201 | 93,303 | 10,055 |
| Software design and architecture | 496 | 360 | 43 | 82,141 | 16,658 |
| Single chip microcomputer principle and App. | 784 | 505 | 88 | 80,977 | - |
| Network programming technology | 1,236 | 362 | 71 | 77,571 | - |
| Digital logic | 1,805 | 348 | 64 | 76,986 | 382 |
| Introduction to computer | 1,801 | 546 | 99 | 63,889 | - |
| Numerical analysis | 610 | 1,111 | 241 | 54,558 | 17,348 |
| Big data management | 217 | 640 | 65 | 51,724 | - |
| Psychology | 296 | 111 | 1 | 50,180 | - |
| Probability and statistic | 557 | 1,054 | 247 | 46,106 | 1,413 |
| Problem solving fundation | 1,037 | 373 | 63 | 38,440 | 323 |
| Artificial intelligence | 582 | 353 | 77 | 28,088 | - |
| Intro to artificial intelligence | 740 | 301 | 58 | 24,879 | 11 |
| Software testing and quality assurance | 454 | 172 | 4 | 24,342 | - |
| Linear algebra | 494 | 420 | 100 | 17,341 | 8,811 |
| PHP programming | 165 | 632 | 153 | 14,462 | 340 |
| Object oriented analysis and design | 265 | 178 | 47 | 11,872 | - |
| Introduction to internet of things | 217 | 291 | 14 | 10,949 | - |
| Microcomputer principle and interface tech. | 422 | 41 | 19 | 10,213 | - |
| Signals and systems | 378 | 38 | 12 | 8,502 | - |
| Calculus | 154 | 357 | 120 | 7,024 | 9,085 |
| Matlab simulation | 249 | 76 | 10 | 6,030 | 6,089 |
| Japanese | 54 | 190 | 19 | 5,375 | - |
| Computer system fundamentals | 67 | 91 | 25 | 3,643 | - |
| Introduction to cloud computing | 104 | 59 | 39 | 3,571 | - |
| Wireless network | 60 | 54 | 26 | 2,242 | - |
| Swift programming | 31 | 102 | 14 | 2,170 | - |
| Data visualization | 87 | 53 | 6 | 2,105 | 513 |
| Fundamentals of analogy electron technique | 36 | 29 | 19 | 1,476 | - |
| Politics | 56 | 29 | 3 | 1,450 | - |
| Tourism | 22 | 30 | 1 | 1,320 | - |
| Software requirement analysis and design | 331 | 21 | 20 | 21 | 5,492 |
| Haskell programming | 98 | 3 | 3 | - | 302 |

# C  Implementation Details

CCLMF is a model-agnostic framework that can be applied to various CD or KT models. Here, we take NCD as an example and showcase the implementation details of CCLMF based on NCD, namely CC-NCD. After pre-training the NCD model in the source course, the student's proficiency representation in the source course can be obtained by extracting the corresponding row from the

matrix $\mathbf{A}^s$ given the student ID $i$:

$$\boldsymbol{u}_i^s = \mathbf{A}_{\text{NCD}}^s[i], \tag{1}$$

where $\mathbf{A}^s$ is the student representation matrix learned by NCD.

In the meta stage, we used a two-layer perceptron (MLP) as the meta network. This meta network then generates a transformation matrix for each student as the personalized mapping function:

$$\boldsymbol{T}_{K^s \times K^t} = \text{MLP}(\boldsymbol{u}_i^s; \theta), \tag{2}$$

where $\theta$ is the parameters of MLP, and $\boldsymbol{T}_{K^s \times K^t}$ is the transformation matrix. $K^s$ and $K^t$ denote the dimensionality of the student proficiency representation in the source and target course respectively. Specifically, the dimensionality of the student representation is determined by the number of knowledge concepts considered.

The transformation matrix $\boldsymbol{T}_{K^s \times K^t}$ is then used to map student proficiency representation to the target course using matrix multiplication:

$$\boldsymbol{u}_i^t = \boldsymbol{u}_i^s \cdot \boldsymbol{T}_{K^s \times K^t}. \tag{3}$$

The final output $\hat{r}_i$ of CC-NCD is formulated as:

$$\hat{r}_i = L(\boldsymbol{Q}_p^t \circ (\boldsymbol{u}_i^t - \boldsymbol{h}^{diff}) \times h^{disc}; \theta_l), \tag{4}$$

where $\boldsymbol{Q}_p^t \in \{0,1\}^{1 \times K^t}$ is the concept relevancy of the problem $p$ in the target course. $\boldsymbol{h}^{diff} \in (0,1)^{1 \times K^t}$, $h^{disc} \in (0,1)$ denotes concept difficulty and problem discrimination learned from the NCD model using data of the target source. $L(\cdot)$ denotes the Linear Layers in NCD which is shown in full paper Figure 5 and $\theta_l$ is the parameters of $L(\cdot)$.

Given the ground truth value $r$ from $\mathcal{R}^t$, all learnable parameters are trained together with the meta network and mapping function by optimizing the cross-entropy loss function as:

$$loss_{CC-NCD} = -\sum_i (r_i \log \hat{r}_i + (1 - r_i) \log (1 - \hat{r}_i)). \tag{5}$$

During the inference stage, given a cold-start student $s_j$ in the target course, we can get the latent proficiency representation in the target course as:

$$\boldsymbol{u}_j^t = \mathbf{A}_{\text{NCD}}^s[j] \cdot \text{MLP}(\mathbf{A}_{\text{NCD}}^s[j]; \theta), \tag{6}$$

which can be utilized to predict the student's performance in the target course via Equation (4).

## D  Baseline Model Details

Cognitive diagnosis models:

**DINA [4, 22]** is a traditional method that is well-suited for binary scoring items, and it can effectively account for student errors due to guessing or slipping.

**IRT [5]** is an important psychological and educational theory rooted in psychometrics, which employs a linear function to model the features of both students and problems.

**MIRT [19]** is a multidimensional extension of IRT, modeling multiple knowledge proficiency.

**NCD [23]** is the first attempt to introduce neural networks for Cognitive Diagnosis, which can model high-order and complex student-problem interaction.

**RCD [6]** models the interactive and structural relations via a multi-layer student-problem-concept relation map and infers students' proficiency through the representations from this map.

Knowledge tracing models:

**DKT [17]** is the first approach applying deep learning to knowledge tracing tasks, making use of the recurrent neural network in the process of modeling students' behavior.

**DKVMN [26]** makes use of a memory network, a static matrix to store all concepts and a dynamic matrix to update students' knowledge states of those concepts.

**SAKT [16]** employs a self-attention mechanism to capture the connections between exercises and student responses.

**AKT [7]** utilizes an attention mechanism to analyze the temporal gap between questions and students' prior interactions to better understand their past engagement.

**GIKT [25]** makes use of a bipartite graph to model the input information, namely problems and concepts, and uses graph convolutional neural network (GCN) to process the data. Then the results were sent to LSTM and get the final prediction.

**SGKT [24]** uses a session graph and during the process of students' answering, a gated graph neural network was set to handle the problem-concept graph.

**PEBG [11]** utilizes pre-training model to get the low-dimensional problem embeddings and models the relation between problems and concepts as a bipartite graph.

## E  Supplementary Experiment Results

We conducted experiments between the selected five courses in Figure 4(a) of the paper. The experimental results are presented in Table 2. We chose *C++ Programming* as the target course due to its wide range of correlation coefficients with other courses. The source courses are marked within brackets and are ranked from lowest to highest correlation coefficient with *C++ Programming*: 0.54 for *Python Programming* (Python), 0.59 for *Data Structure and Algorithm Analysis* (DS), 0.64 for i (Java), 0.79 for *C Programming* (C). To simulate cold-start scenarios, we sampled 5% of each student's response logs in *C++ Programming* to form the target course.

From Table 2, we can observe that CCLMF achieves a certain improvement of the two models in all source courses. Notably, the results of the NCD model reveal that the extent of model improvement is related to the correlation coefficient between the source course and the target course. The source course with the highest correlation coefficient (0.79 for *C Programming*) exhibits the most significant improvement, while the source course with the weakest correlation coefficient (0.54 for *Python Programming*) demonstrates relatively less improvement.

Table 2: CCLMF results on MIRT and NCD, taking *C++ Programming* as the target course. Source courses are marked within brackets.

| Metrics | Model | no dropout | 10% dropout | 20% dropout | 30% dropout | 40% dropout | 50% dropout |
|---|---|---|---|---|---|---|---|
| AUC | MIRT | 0.6272 | 0.6218 | 0.6157 | 0.6124 | 0.6051 | 0.6057 |
| | CC-MIRT (Python) | **0.7150 (+0.0878)** | **0.7123 (+0.0905)** | **0.7102 (+0.0945)** | 0.6864 (+0.0740) | 0.6801 (+0.0750) | 0.6716 (+0.0659) |
| | CC-MIRT (DS) | 0.6912 (+0.0640) | 0.6933 (+0.0715) | 0.6904 (+0.0747) | **0.7012 (+0.0888)** | **0.7042 (+0.0991)** | **0.6990 (+0.0933)** |
| | CC-MIRT (Java) | 0.7132 (+0.0860) | 0.7093 (+0.0875) | 0.7065 (+0.0908) | 0.7004 (+0.0880) | 0.6975 (+0.0924) | 0.6890 (+0.0833) |
| | CC-MIRT (C) | 0.6996 (+0.0724) | 0.7021 (+0.0803) | 0.6935 (+0.0778) | 0.6912 (+0.0788) | 0.6870 (+0.0819) | 0.6768 (+0.0711) |
| ACC | MIRT | 0.7493 | 0.7479 | 0.7479 | 0.7471 | 0.6825 | 0.6860 |
| | CC-MIRT (Python) | **0.7738 (+0.0245)** | **0.7721 (+0.0242)** | **0.7714 (+0.0235)** | 0.7657 (+0.0186) | 0.7637 (+0.0812) | 0.7606 (+0.0746) |
| | CC-MIRT (DS) | 0.7668 (+0.0175) | 0.7681 (+0.0202) | 0.7665 (+0.0186) | **0.7694 (+0.0223)** | **0.7706 (+0.0881)** | **0.7685 (+0.0825)** |
| | CC-MIRT (Java) | 0.7719 (+0.0226) | 0.7707 (+0.0228) | 0.7706 (+0.0227) | 0.7672 (+0.0201) | 0.7671 (+0.0846) | 0.7641 (+0.0781) |
| | CC-MIRT (C) | 0.7703 (+0.0210) | 0.7719 (+0.0240) | 0.7695 (+0.0216) | 0.7683 (+0.0212) | 0.7671 (+0.0846) | 0.7643 (+0.0783) |
| RMSE | MIRT | 0.4919 | 0.4935 | 0.4935 | 0.4931 | 0.511 | 0.5106 |
| | CC-MIRT (Python) | **0.4009 (-0.0909)** | **0.4016 (-0.0919)** | **0.4022 (-0.0913)** | 0.4104 (-0.0827) | 0.4118 (-0.0992) | 0.4151 (-0.0955) |
| | CC-MIRT (DS) | 0.4089 (-0.0830) | 0.4074 (-0.0861) | 0.4086 (-0.0849) | **0.4052 (-0.0879)** | **0.4035 (-0.1075)** | **0.4054 (-0.1052)** |
| | CC-MIRT (Java) | 0.4021 (-0.0898) | 0.4027 (-0.0908) | 0.4035 (-0.0900) | 0.4055 (-0.0876) | 0.4062 (-0.1048) | 0.4085 (-0.1021) |
| | CC-MIRT (C) | 0.4050 (-0.0869) | 0.4039 (-0.0896) | 0.4062 (-0.0873) | 0.4071 (-0.0860) | 0.4085 (-0.1025) | 0.4112 (-0.0994) |
| AUC | NCD | 0.6981 | 0.6960 | 0.6926 | 0.6873 | 0.6846 | 0.6791 |
| | CC-NCD (Python) | 0.7008 (+0.0028) | 0.6979 (+0.0019) | 0.6943 (+0.0017) | 0.6931 (+0.0058) | 0.6873 (+0.0027) | 0.6807 (+0.0016) |
| | CC-NCD (DS) | 0.7225 (+0.0244) | 0.7189 (+0.0229) | 0.7164 (+0.0238) | 0.7128 (+0.0255) | 0.7078 (+0.0232) | 0.6997 (+0.0206) |
| | CC-NCD (Java) | 0.7154 (+0.0173) | 0.7123 (+0.0163) | 0.7079 (+0.0153) | 0.7058 (+0.0185) | 0.6989 (+0.0143) | 0.6907 (+0.0116) |
| | CC-NCD (C) | **0.7663 (+0.0682)** | **0.7627 (+0.0667)** | **0.7598 (+0.0672)** | **0.7541 (+0.0668)** | **0.7486 (+0.0640)** | **0.7423 (+0.0632)** |
| ACC | NCD | 0.7619 | 0.7602 | 0.7616 | 0.7552 | 0.7556 | 0.7558 |
| | CC-NCD (Python) | 0.7693 (+0.0074) | 0.7675 (+0.0073) | 0.7666 (+0.0050) | 0.7674 (+0.0122) | 0.7668 (+0.0112) | 0.7668 (+0.0110) |
| | CC-NCD (DS) | 0.7747 (+0.0128) | 0.7702 (+0.0100) | 0.7677 (+0.0061) | 0.7673 (+0.0121) | 0.7675 (+0.0119) | 0.7671 (+0.0113) |
| | CC-NCD (Java) | 0.7661 (+0.0043) | 0.7695 (+0.0093) | 0.7687 (+0.0071) | 0.7667 (+0.0115) | 0.7668 (+0.0112) | 0.7660 (+0.0102) |
| | CC-NCD (C) | **0.7854 (+0.0235)** | **0.7875 (+0.0273)** | **0.7833 (+0.0217)** | **0.7759 (+0.0207)** | **0.7734 (+0.0178)** | **0.7710 (+0.0152)** |
| RMSE | NCD | 0.4116 | 0.4124 | 0.4115 | 0.4174 | 0.4164 | 0.4156 |
| | CC-NCD (Python) | 0.4102 (-0.0014) | 0.4094 (-0.0030) | 0.4111 (-0.0004) | 0.4115 (-0.0059) | 0.4123 (-0.0041) | 0.4130 (-0.0026) |
| | CC-NCD (DS) | 0.4018 (-0.0098) | 0.4059 (-0.0065) | 0.4081 (-0.0034) | 0.4133 (-0.0041) | 0.4123 (-0.0041) | 0.4181 (0.0025) |
| | CC-NCD (Java) | 0.4088 (-0.0028) | 0.4066 (-0.0058) | 0.4075 (-0.0040) | 0.4097 (-0.0077) | 0.4104 (-0.0060) | 0.4130 (-0.0026) |
| | CC-NCD (C) | **0.3877 (-0.0239)** | **0.3881 (-0.0243)** | **0.3913 (-0.0202)** | **0.3956 (-0.0218)** | **0.3967 (-0.0197)** | **0.4028 (-0.0128)** |

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
