# OpenReview forum: "PTADisc: A Cross-Course Dataset Supporting Personalized Learning in Cold-Start Scenarios"
_NeurIPS.cc/2023/Track/Datasets_and_Benchmarks — NeurIPS 2023 Datasets and Benchmarks Poster_

### Official Review · Reviewer_G8A9 · 2023-07-17
**Review for PTADisc: A Diverse, Immense, Student-Centered and Cross-Course Dataset for Personalized Learning**

**Rating:** 6
**Confidence:** 2

**Strengths:**

- The paper has stated their motivation clearly and gives detailed argument of why adding cross-course component is necessary
- They have tackled quite a few challenges in this domain, such as data coverage, concept annotations, and cross-course information. The efforts to get rid of these problem in a large scale data is appreciated

**Additional Feedback:**

N/A

**Clarity:**

- How to make sense Figure 4 (a). Given the cross-course correlation coefficients are relatively low (except for C and C++), what are the intuition behind cross-course information being helpful?
- In Table 3, what are intuitions to have multiple dropout ratio of the dataset? It's unclear why it's important to incorporate them here

**Correctness:**

- In the cold start simulation, they merely compare the course Python Programming to Java Programming.
- First this is only one simulation, so how can they demonstrate the overall cold-start capability of their benchmark?
- Second these two classes have some similarities, why not use the classes with content that are far more different?

**Documentation:**

- The data set is not clearly documented. Furthermore, the data is not sharable according to the authors and they only aim to share the extracted features of the dataset which defeats the purpose of this paper. Couldn't they scrub personal information and protect the privacy similar to clinical dataset?

**Ethics:**

The dataset contains personal information from the students. However, according to the authors, they took some actions towards this end.

**Limitations:**

- From Figure 3, I can't infer what's the raw data and thus the final form of data. There needs more explanation on the raw data they sourced from the PTA platform
- In section 3.2.3, if knowledge tracing and cognitive diagnosis dataset only differ on submitting time, why make them as two separate tasks?

**Opportunities For Improvement:**

- The authors haven't stated clearly how this data can spur other ML researcher to use it. The data type and format is vaguely described and not clearly discussed in introduction. In fact, after reading introduction, I'm still confused what's the proposed dataset's utility in real life
- They refer to the website of https://pintia.cn/ with the hope that the reader would get familiarized with what's PTA but this is not suitable since it's the author's responsibility to describe the problem setting clearly

**Relation To Prior Work:**

The authors discussed the comparison between their work and prior works.

**Summary And Contributions:**

This paper presents a new dataset for evaluation of student's personalized learning outcome, mainly focusing on cognitive diagnosis and knowledge tracing. The contribution lies in adding cross-course component in the study. The dataset is benchmarked on a pipeline (CCLMF) which is model agnostic, to evaluate the student's proficiency across courses as well as tackling cold-start scenario.

---

> ### Author Response · Authors · 2023-08-15
> **Individual Response to Reviewer G8A9 (Part A)**
>
> We thank reviewer G8A9 for the valuable feedback and recommendations for improving the manuscript. In light of your feedback, we addressed the concerns as described below.
>
> ### Response to the “Opportunities For Improvement” section:
>
> > 1. How this data can spur other ML researcher to use it.
>
> Thank you for your valuable suggestion. We have provided additional explanations regarding the applications of PTADisc in General Response 3. In general, the construction of the PTADisc dataset serves as a fundamental data resource for the development of various intelligent educational algorithms/applications. These algorithms/applications play a crucial role in aiding teachers to gain a deeper understanding of students' knowledge states and facilitate students in devising effective learning paths, thereby enhancing their overall learning efficiency.
>
> For the data format, please refer to General Response 2 for detailed information. Additionally, we have included examples of course, problem, and response logs in Tables 1, 2, and 3 of our revised paper. These examples are intended to provide a clearer understanding of the data format. For more detailed information of the format of the data, we provide a comprehensive description on our [github page](https://github.com/wahr0411/PTADisc).
>
> > 2. Lack of introduction to https://pintia.cn/.
>
> Thanks for pointing this out. We have supplemented the related description at the beginning of Section 3 in our revised paper.
>
> PTADisc is sourced from PTA, an online learning platform developed by Hangzhou PAT Education Technology Co., Ltd. PTA is an automatic program evaluation and open teaching assistance platform for universities and society. Given the close collaboration between PTA and universities, it is common for students to concurrently pursue a series of courses that align with their training program. Up to July 2023, PTA has attracted over $1,000$ organizations, $9,000$ teachers and $3,900,000$ users and provides a problem bank of over $290,000$ problems referenced by course problem sets and exams. The highlight of PTA is that it covers a significant amount of students enrolled in multiple courses. This feature perfectly meets the need to conduct cross-course research and mine student characteristics between courses.
>
> ### Response to the “Limitations” section:
> > 1. What's the raw data and thus the final form of data.
>
> Thanks for your requirements regarding the data format. We have included examples of course, problem, and response logs in Tables 1, 2, and 3 of our revised paper. For a detailed data format description, please refer to General Response 2. And further comprehensive documentation can be found on our [github page](https://github.com/wahr0411/PTADisc).
>
> Actually, there is no significant difference between the raw data and the structured data in its final form. During the processing of raw data, the main emphasis was on safeguarding privacy, eliminating redundant information, and addressing unannotated data, all while ensuring the overall information was preserved. The structured data includes all pertinent data that could be retained, and the details presented in Table 5 of our revised paper are openly accessible.
>
> > 2. Why make CD and KT as two separate tasks?
>
> We summarize the differences between KT and CD from the task objective and modeling approach:
>
> - Task Objective:
>   - KT considers the temporal order of students’ sequential exercising records, where the student's knowledge state keeps changing. The goal is to predict whether the next question can be answered correctly, taking into account the sequential dependencies and patterns within the data. This task can be analogized to sequential recommendation.
>
>   - CD treats students' exercising records as unordered, where the student's knowledge state is considered static. The goal is to identify a student's specific cognitive strengths and weaknesses in relation to predefined skills during a period of time. This task can be analogized to the traditional recommendation.
>
> - Modeling Approach:
>
>   - KT models aim to capture the knowledge acquisition process of students over time. These models utilize sequential data, such as the order of students' interactions with learning materials, to estimate the student's knowledge state at each point in time. Popular knowledge tracing models include Bayesian Knowledge Tracing (BKT) [1] and Deep Knowledge Tracing (DKT) [2], which use machine learning techniques like hidden Markov models or recurrent neural networks.
>
>   - CD models are designed to capture the relationships between question embeddings and latent cognitive attributes, with a greater emphasis on the student's learning status and their mastery of the questions (whether they can answer correctly). Mainstream CD models include conventional methods like MIRT [3] which leverages linear functions, and the deep learning method NCD [4] which employs neural networks to extract features from a higher-dimensional space.

---

> > ### Author Response · Authors · 2023-08-15
> > **Individual Response to Reviewer G8A9 (Part B)**
> >
> > ### Response to the “Correctness” section:
> > > In the cold start simulation, they merely compare the course Python Programming to Java Programming. First this is only one simulation, so how can they demonstrate the overall cold-start capability of their benchmark? Second these two classes have some similarities, why not use the classes with content that are far more different?
> >
> > First, we'd like to explain why we chose *Python Programming* and *Java Programming*. As stated in [5], cross-domain analysis research requires content-level or user-level relevance between the domains being studied. Similarly, in our cross-course study, it is vital to have a certain level of similarity between the selected courses in order to effectively address the cold start problem. This similarity can manifest in shared students and correlations in student performances across courses. Such similarity serves as the foundation for extracting beneficial cross-domain mapping functions that can address the cold start problem. The correlation coefficient between students' performance in *Python Programming* and *Java Programming* is $0.65$, indicating a relatively high correlation. This high correlation makes these two courses well-suited for addressing cold-start problems. And we also supplemented this description in Section 4.3 of our revised paper.
> >
> > Specifically, within the discipline of computer science, the degree of correlation in students' performance across different courses varies. As you suggested, we have conducted additional experiments between the selected five courses in Figure 4(a), and the results are presented in Supplementary Material E. The results can be seen [here](https://github.com/wahr0411/PTADisc/blob/main/img/supplementary%20experiment.png).
> >
> > For the supplementary experiments, we chose *C++ Programming* as the target course due to its wide range of correlation coefficients with other courses. The source courses are marked within brackets and are ranked from lowest to highest correlation coefficient with *C++ Programming*: $0.54$ for *Python Programming* (Python), $0.59$ for *Data Structure and Algorithm Analysis* (DS), $0.64$ for *Java Programming* (Java), $0.79$ for *C Programming* (C). To simulate cold-start scenarios, we sampled 5% of each student’s response logs in *C++ Programming* to form the target course. From the table, we can observe that CCLMF achieves a certain improvement over the two models in all source courses. Notably, the results of the NCD model reveal that the extent of model improvement is related to the correlation coefficient between the source course and the target course. The source course with the highest correlation coefficient ($0.79$ for *C Programming*) exhibits the most significant improvement, while the source course with the weakest correlation coefficient ($0.54$ for *Python Programming*) demonstrates relatively less improvement.
> >
> > ### Response to the “Clarity” section:
> >
> > > 1. How to make sense Figure 4 (a)?
> >
> > First of all, we'd like to provide a supplementary description of the calculation method of the correlation coefficient. We considered approximately 30,000 students who enrolled in all five courses and calculated their scoring rates for non-programming problems by dividing their final scores achieved by the total score. Subsequently, the correlation coefficient was computed based on the scoring rate arrays of the students across these courses.
> >
> > According to [6], correlation coefficients with values between 0.5 and 0.7 indicate a moderate level of correlation, and between 0.7 and 0.9 indicate a high level of correlation  between variables. In Figure 4(a), the correlation coefficients fall within these ranges, suggesting a certain degree of correlation between the courses. This implies that students' mastery of knowledge concepts in these courses is positively related. Based on this observation, we hypothesized that this positive correlation could be leveraged to address the cold-start problem, leading us to design the CCLMF method. The experimental results not only confirmed the effectiveness of utilizing cross-course information in improving performance in cold-start scenarios but also demonstrated a greater level of improvement with higher correlation coefficients.
> >
> > > 2. What are intuitions to have multiple dropout ratio of the dataset?
> >
> > The different dropout ratios employed in Table 3 are aimed at simulating various levels of data sparsity and cold-start situations. This approach allows us to effectively showcase the performance enhancement of our proposed CCLMF method in cognitive diagnosis tasks across different data sparsity and cold-start conditions.

---

> > ### Comment · Reviewer_G8A9 · 2023-08-21
> >
> > Thanks for the clarification on KT and CD. We can argue in terms of real world utility they are different methods to assess the learning results. However as a dataset paper, the work to create a set of CD labels and then add timestamps to them to create KT task seems trivial. To enrich any machine learning model, the task should be diverse in nature, and that reflects on how the labels are created to meet more diverse needs in real life.

---

> > > ### Author Response · Authors · 2023-08-23
> > > **Reply to Reviewer G8A9 (Part A)**
> > >
> > > We thank reviewer G8A9 for the insightful comments. We hope that our responses will address your concerns.
> > >
> > > **We would like to politely emphasize that the distinctive contribution of PTADisc lies in providing cross-course information, rather than simultaneously supporting both CD and KT tasks.** The primary objective behind the construction of this dataset was to investigate the relationship of students' performance across different courses, as we believe that cross-course analysis holds significant value in addressing the cold-start problem. Most existing datasets are constructed within a single course (e.g [ASSIST09](https://sites.google.com/site/assistmentsdata/home/2009-2010-assistment-data) in Math, and EdNet [1] in English). Despite MOOCCubeX [2] consists of $4,216$ courses, there exist no students with learning records in multiple courses. The lack of cross-course dataset motivated us to construct one to support our research, serving as the primary motivation behind our work. Therefore, leveraging information from the PTA platform, we developed the PTADisc dataset, which is the first dataset to include a set of students with learning records across multiple courses simultaneously.
> > >
> > > Below, **we provide a concise overview of the dataset construction process to facilitate a better understanding of the contribution of our dataset.** Initially, we collected students' learning behavior data and other relevant metadata from approximately 74 courses available on the PTA platform. Subsequently, we performed statistical analysis to identify students who had enrolled in multiple courses. By doing so, we determined the top five most frequent combinations of courses and aligned the students across these specific course sets. To ensure dataset quality, we further eliminated students with less than 10 response logs for any of the five courses. As a result, we obtained a set of 29,454 students. Every student in the set had taken all five courses with exercise records exceeding 10 in each course.
> > >
> > > Leveraging this subset of cross-course data, we introduced CCLMF, **an approach aimed at mitigating the challenges associated with the cold-start problem in educational cognitive tasks**. In this context, the cold-start problem refers to the difficulty of accurately predicting students' performance in new courses when there is insufficient or no historical data available for students in those courses. Addressing this problem is crucial for enhancing the effectiveness and user experience of computer-aided education systems. CCLMF leverages student latent proficiency relationships between courses to transfer knowledge from courses with sufficient exercise records, thereby improving performance in the target course. The experimental results verify the effectiveness of CCLMF with an average improvement of 4.2% on AUC.
> > >
> > > Meanwhile, our dataset also offers a wealth of additional information (refer to Table 5) that can be utilized for various tasks. In fact, most of the existing CD and KT methods primarily use only the problem IDs and student IDs. Differently, our dataset offers more comprehensive data, such as knowledge concept correlations, problem sets, and programming records, which facilitate a deeper analysis of student performance in these two tasks in the future. Specifically, PTADisc contains a significant number of programming response logs, which can support the **programming knowledge tracing task**. This task involves utilizing students' sequential submission records for programming problems, including details such as the code submitted in each attempt, the programming language employed, and additional information like running time and memory consumption. It is important to highlight that students often make multiple submissions and revisions for a single programming question. The main objective of this task is to evaluate students' proficiency in programming skills and predict their potential scores in relation to each testing point for their subsequent submissions.
> > >
> > > Besides KT and CD, **the information provided by PTADisc can also support other tasks**. For example, prerequisite discovery [3], which refers to the task of identifying and establishing the sequence or order in which concepts or topics should be learned or presented, ensuring that foundational concepts are understood before more advanced ones. Computerized adaptive testing [4], which is an emerging testing format in many standardized examinations, aims to rapidly and accurately diagnose a candidate's level of knowledge mastery through personalized test items/exercises. Educational recommendation [5], which is capable of making appropriate learning suggestions to students, considering the interactions between students and questions.

---

> > > ### Author Response · Authors · 2023-08-23
> > > **Reply to Reviewer G8A9 (Part B)**
> > >
> > > **Apologies for any confusion regarding CD and KT**. The experiments conducted in Section 5 primarily serve to showcase the reliability and generalizability of our dataset across different existing methods for these two classical tasks. This particular analysis is not directly related to the core contribution of our work. Both the CD and KT tasks, as well as our proposed CCLMF method, revolve around diagnosing students' level of knowledge mastery. However, the cross-domain study we introduced necessitates addressing data disparities and ensuring feature alignment between courses, while simultaneously enabling the analysis of students' learning performance across multiple courses
> > >
> > > **We would like to politely ask the reviewer to re-evaluate our work.** We are open to further discussion with the reviewer regarding the revised paper. Thanks again.
> > >
> > > **Reference**
> > >
> > > [1] Choi, Youngduck, et al. "Ednet: A large-scale hierarchical dataset in education." Artificial Intelligence in Education: 21st International Conference, AIED 2020, Ifrane, Morocco, July 6–10, 2020, Proceedings, Part II 21. Springer International Publishing, 2020.
> > >
> > > [2] Yu, Jifan, et al. "MOOCCubeX: a large knowledge-centered repository for adaptive learning in MOOCs." Proceedings of the 30th ACM International Conference on Information & Knowledge Management. 2021.
> > >
> > > [3] Pan, Liangming, et al. "Prerequisite relation learning for concepts in moocs." *Proceedings of the 55th Annual Meeting of the Association for Computational Linguistics (Volume 1: Long Papers)*. 2017.
> > >
> > > [4] Bi, Haoyang, et al. "Quality meets diversity: A model-agnostic framework for computerized adaptive testing." 2020 IEEE International Conference on Data Mining (ICDM). IEEE, 2020.
> > >
> > > [5] Huang, Zhenya, et al. "Exploring multi-objective exercise recommendations in online education systems." *Proceedings of the 28th ACM International Conference on Information and Knowledge Management*. 2019.

---

> > > > ### Comment · Reviewer_G8A9 · 2023-08-28
> > > >
> > > > Thanks authors for detailed responses. The discussion above cleared up many confusions. However in terms of CD and KT, if the benchmarks conducted are not necessarily related to the core contribution of the work, then what's the point incorporating them. Nonetheless, I believe by examining the raw data form (which is the question I brought up earlier), more useful and relevant labels/tasks can be derived. I suppose the data release will eventually benefit the community but encourage the authors to continue to think about label definitions and the implications of them in real world. And as such, I would raise the score.

---

> > > > > ### Author Response · Authors · 2023-08-28
> > > > >
> > > > > We are very pleased to hear that our response has addressed your questions and concerns.
> > > > >
> > > > > We acknowledge your concern regarding the inclusion of benchmarks that may not directly align with the core contribution of our work. In order to provide a clearer understanding, we will expand upon the explanation of how CD and KT tasks form the basis of our cross-course investigation.
> > > > >
> > > > > Moreover, we will continue to explore the possibilities and consider refining our label definitions accordingly.
> > > > >
> > > > > We sincerely appreciate the time and effort you have dedicated to reviewing our work. Thank you once again for your valuable feedback.

---

> ### Author Response · Authors · 2023-08-15
> **Individual Response to Reviewer G8A9 (Part C)**
>
> ### Response to the “Documentation” section:
> > 1. The data set is not clearly documented.
>
> As anticipated, we have indeed made all the data publicly available, as clarified in the Response to the “Limitations” section, while also ensuring necessary privacy protection measures were in place. More information regarding the specific types of retained data and detailed data formats can be found in General Response 2.
>
> **Reference**
>
> [1] Kobsa, Alfred. "User modeling and user-adapted interaction." Conference companion on Human factors in computing systems. 1994.
>
> [2] Piech, Chris, et al. "Deep knowledge tracing." Advances in neural information processing systems 28 (2015).
>
> [3] Reckase, Mark D. "18 Multidimensional Item Response Theory." Handbook of statistics 26 (2006): 607-642.
>
> [4] Wang, Fei, et al. "Neural cognitive diagnosis for intelligent education systems." Proceedings of the AAAI conference on artificial intelligence. Vol. 34. No. 04. 2020.
>
> [5] Zhu, Feng, et al. "Cross-domain recommendation: challenges, progress, and prospects." arXiv preprint arXiv:2103.01696 (2021).
>
> [6] [correlation coefficie](https://www.andrews.edu/~calkins/math/edrm611/edrm05.htm). Retrieved 8 August 2023

---

### Official Review · Reviewer_SZNR · 2023-07-20
**A new treasure-trove for educational modeling**

**Rating:** 8
**Confidence:** 4
**Correctness:** Cannot be estimated.

**Strengths:**

First, it is a very big new dataset, with cross-course information.
Second, a model is presented for utilizing information across courses for proficiency prediction, which is can be useful in 'cold-start' cases, i.e. when not much is known yet about student performance in the target course.

**Additional Feedback:**

N/A

**Clarity:**

The paper is written in 'bird's eye' overview manner. It gives a very high-level overview of the data, but no details. For example it mentions knowledge concepts and their hierarchical relations, but no examples are given, an a hint in Fig.1 is a very small schematic generic diagram.  This might be OK for people already working in this area of research, but can be very unclear to everyone else.
More information, maybe with examples, is need about the kind of concepts included in the dataset.


**Documentation:**

Considerable documentation is provided in the supplementary material.

**Ethics:**

no ethical concerns

**Limitations:**

Lack of textual content of the problems in the dataset – as stated by the authors.

**Opportunities For Improvement:**

N/A

**Relation To Prior Work:**

Prior work is well described and cited.

**Summary And Contributions:**

This paper presents a new dataset of educational records. It is a huge data set, covering many courses and more than a million students. The data is university-level courses, mostly programming. A novel aspect of the dataset is that information for many students is available for more than one course.
The authors demonstrate the advantage of such a dataset is for predicting student proficiency across courses, by utilizing student data from one course to predict the student's proficiency in another course.

---

> ### Author Response · Authors · 2023-08-15
> **Individual Response to Reviewer SZNR**
>
> We thank the reviewer SZNR for their constructive comments and suggestions. Following your feedback, we have revised the manuscript accordingly, and the detailed response is presented below.
>
> ### Response to the “Clarity” section:
>
> > 1. The paper is written in 'bird's eye' overview manner. It gives a very high-level overview of the data, but no details. For example it mentions knowledge concepts and their hierarchical relations, but no examples are given, an a hint in Fig.1 is a very small schematic generic diagram. This might be OK for people already working in this area of research, but can be very unclear to everyone else. More information, maybe with examples, is need about the kind of concepts included in the dataset.
>
> Thanks for your requirements regarding the data format and examples. We have included examples of course, problem, and response logs in Tables 1, 2, and 3 of our revised paper. For a detailed data format description, please refer to General Response 2. And further comprehensive documentation can be found on our [github page](https://github.com/wahr0411/PTADisc).
>
> To illustrate the knowledge concepts and their hierarchical relations, we present an example table showcasing the concepts in Python Programming. The course ID assigned to Python Programming is C_9088, which also serves as the root node of the concept tree.
>
> |Concept ID|Concept Name|Concept Parent ID|
> |-------------|-----------------|--------------------- |
> | C\_1568    | Loop              | C\_9088                |
> | C\_9472    | Function        | C\_9088                |
> | C\_2592    | Break             | C\_1568                |
> | C\_7488    | Continue        | C\_1568                |

---

### Official Review · Reviewer_ieLM · 2023-07-21
**A valuable educational dataset, not only fitting the needs of Cross-Course analysis, but also supporting a wider range of applications**

**Rating:** 9
**Confidence:** 4
**Clarity:** The paper is well written.

**Strengths:**

1. The proposed dataset is immense and valuable, benefiting  data mining community and the pedagogy community.
2. PTADisc is the first dataset with cross-course information. The analysis of student latent proficiency relationships between courses makes a contribution to personalized learning.
3. PTADisc is currently the largest educational dataset. It contains extensive concept-related information and detailed records of student behaviors, which is well-suited for diagnostic tasks.
4. Experimental results demonstrate the effectiveness of the proposed method and the generalizability of the dataset. The paper is well-written and addresses important concerns, such as ethics issue.


**Additional Feedback:**

N/A

**Correctness:**

The claims in this paper are correct, and the datasets are constructed in a sound way. The evaluation methods and experiment design are correct. The dataset and source code are well organized , hence it should be straightforward for another group to utilize this dataset.

**Documentation:**

The dataset and code are well documented.

**Ethics:**

The dataset submission mentioned privacy protection in 3.1, indicating that the creators have considered the ethical implications of the data.

**Limitations:**

The author has declared the limitation of the dataset in conclusion section. Hope they can release the text features in the future, which is important for knowledge tracing and cognitive diagnosis tasks.

**Opportunities For Improvement:**

In addition to student behaviors, the dataset also contains concept-problem relation information and programming records. These additional types of data provide a multitude of research opportunities, but the definition about these tasks are not well discussed and summarized in the paper. To address this, it is recommended to include a subsection that outlines the potential applications of the dataset across various tasks.

**Relation To Prior Work:**

The author has discussed the distinctions between this work and others, as stated in lines 31-40. The primary difference lies in the emphasis on cross-course learning, which is the fundamental differentiating factor for this dataset when compared to others.

**Summary And Contributions:**

This paper introduces a novel educational data mining dataset named PTADisc, specifically designed to facilitate diagnostic tasks in personalized learning. With an extensive volume of student behavior information (over 1.5 million students and 680 million student exercise response logs), PTADisc not only supports a wide range of educational data mining tasks, but also holds potential value for other disciplines such as cognitive psychology. PTADisc stands out as the first dataset to incorporate cross-course information, facilitating the analysis of student behaviors across multiple courses. This dataset includes around 30k students who are simultaneously enrolled in five courses, providing valuable insights into their learning patterns across diverse educational contexts.

This paper also propose a model-agnostic cross-course learner model framework. The evaluation results demonstrate the effectiveness of CCLMF in enhancing diagnostic tasks in cold-start scenarios. The authors also report the performance on cognitive diagnosis and knowledge tracing, demonstrating the proposed dataset can support a wide scope of research.

As a resource paper, the authors have addressed user privacy concerns and implemented measures to ensure data security. The paper is well-written, and the experiments conducted are thorough.

---

> ### Author Response · Authors · 2023-08-15
> **Individual Response to Reviewer ieLM**
>
> We thank the reviewer ieLM for their comprehensive feedback and suggestions. In response, we have made pertinent updates to the manuscript. And a response for your comment is presented below.
>
> ### Response to the “Opportunities For Improvement” section:
>
> > 1. In addition to student behaviors, the dataset also contains concept-problem relation information and programming records. These additional types of data provide a multitude of research opportunities, but the definition about these tasks are not well discussed and summarized in the paper. To address this, it is recommended to include a subsection that outlines the potential applications of the dataset across various tasks.
>
> Thanks for pointing this out. We have supplemented the potential applications of PTADisc in Section 3.5 in our revised paper.
>
> PTADisc not only supports existing CD and KT methods but also supports:
>
> (1) Prerequisite discovery [1], which refers to the task of identifying and establishing the sequence or order in which concepts or topics should be learned or presented, ensuring that foundational concepts are understood before more advanced ones.
>
> (2) Computerized adaptive testing [2], which is an emerging testing format in many standardized examinations, aims to rapidly and accurately diagnose a candidate's level of knowledge mastery through personalized test items/exercises.
>
> (3) Educational recommendation [3], which is capable of making appropriate learning suggestions to students,  considering the interactions between students and questions.
>
> (4) Cross-course research. PTADisc provides cross-course information, enabling the study of how students perform in different classes. We have conducted a cross-course study addressing the cold-start issue in Section 4.
>
> With various information provided, PTADisc also has the potential to support the following research directions and opportunities:
>
> (1) It provides non-binary student performance data. Most existing CD and KT methods tackle this as binary classification (wrong/right answer). Non-binary grades allow for more regression-focused investigations.
>
> (2) It provides information on different problem types, aiding research of CD and KT.
>
> (3) It provides data on problem difficulty, enabling to study how the level of difficulty of exercises relates to students' final learning outcomes.
>
> (4) It provides problem set specifics, like submission time, enabling modeling of students' learning habits based on when they submit their work.
>
> (5) It provides information on student groups, facilitating the assessment of teaching quality within classes and group-level educational analysis.
>
> (6) It provides information on programming exercises, including detailed code submissions and records, enabling in-depth research into programming-related studies.
>
> **Reference**
>
> [1] Pan, Liangming, et al. "Prerequisite relation learning for concepts in moocs." Proceedings of the 55th Annual Meeting of the Association for Computational Linguistics (Volume 1: Long Papers). 2017.
>
> [2] Bi, Haoyang, et al. "Quality meets diversity: A model-agnostic framework for computerized adaptive testing." 2020 IEEE International Conference on Data Mining (ICDM). IEEE, 2020.
>
> [3] Huang, Zhenya, et al. "Exploring multi-objective exercise recommendations in online education systems." Proceedings of the 28th ACM International Conference on Information and Knowledge Management. 2019.

---

> > ### Comment · Reviewer_ieLM · 2023-08-22
> >
> > Thank you for providing a thorough explanation regarding the potential application of PTADisc beyond course-course analysis. I am pleased to note that your response and the revised paper have effectively addressed my concerns.

---

### Official Review · Reviewer_ToYw · 2023-07-21
**A comprehensive and important dataset for AI for Education is proposed.**

**Rating:** 8
**Confidence:** 4
**Correctness:** Yes.
**Clarity:** Yes.

**Strengths:**

1. Reasonable data construction and solid experiment results for benchmarking.
2. The authors investigated a new task of cross-course diagnostic task, which has never been explored in the area of AI for education.
3. PTADisc is a unique dataset that evaluates student responses using a scoring ratio system rather than binary values.

**Additional Feedback:**

The authors should pay attention to the order of figures and tables. We usually place them according to the order of they are referred in the text.

The authors should refine figure 4 (1) by enlarging the text font size.

The authors should give more motivation explanation for using the meta learning strategy.

The authors have addressed all my concerns raised in my review.

**Documentation:**

Yes.

**Ethics:**

No.

**Limitations:**

The authors list the limitation of their work in the paper, which is that currently the original content text of the problems cannot be shared currently due to copyright constraints. They will address them by extracting text features in the future.

**Opportunities For Improvement:**

Lack some analysis and examples on students’ behavior patterns across difference courses. The current analysis and examples provided in the paper focus primarily on students' behavior patterns within the selected five courses, which belong to a similar domain. It would be better to further explore the relationships among courses in different disciplines.

**Relation To Prior Work:**

Yes, clearly discussed and well-positioned.

**Summary And Contributions:**

This paper introduces an open-source educational dataset PTADisc, containing 680 million student response logs. The dataset can be readily utilized for cognitive diagnosis and knowledge tracing tasks. Specially, the dataset can support cross-course studies. In addition, the authors developed a model-agnostic Cross-Course Learner Modeling Framework (CCLMF) which utilizes relationships between students’ proficiency across courses to alleviate the difficulty of diagnosing student knowledge state in cold-start scenarios. Moreover, the model is agnostic and can be integrated with various diagnostic models. The authors provide in-depth analysis about the dataset. Overall, this paper is well structured, easy to follow and critical for AI for education research.

---

> ### Author Response · Authors · 2023-08-15
> **Individual Response to Reviewer ToYw**
>
> We thank reviewer ToYw for their valuable comment and recommendations. Based on your feedback, we have incorporated necessary revisions to the manuscript and provided a detailed response below.
>
> ### Response to the “Opportunities For Improvement” section:
>
> > 1. Lack some analysis and examples on students’ behavior patterns across difference courses. The current analysis and examples provided in the paper focus primarily on students' behavior patterns within the selected five courses, which belong to a similar domain. It would be better to further explore the relationships among courses in different disciplines.
>
> The relationships of students' behavior patterns across different courses are reflected by the correlation coefficient which is computed based on the scoring rate arrays of the students between courses. A positive correlation coefficient (typically greater than 0.5) between courses A and B indicates that as a student performs better in course A, their performance in course B also improves; conversely, a negative coefficient (typically less than -0.5) implies the opposite. The larger the absolute value of the correlation coefficient, the stronger the correlation between the courses. A correlation between -0.5 and 0.5 indicates a weak relationship.
>
> In addition to the examples in Figure 4(a), we have supplemented more correlation relationships between courses in different disciplines. These courses encompass not computer-related courses but also mathematics courses and social science courses, including *Python Programming*, *Database Principle*, *Operating System*, *Discrete Mathematics and Application*, *Linear Algebra*, *English*, and *Literature and History*. The correlation coefficient of students' performance is 0.60 between *Database Principle* and *Discrete Mathematics and Application*, 0.46 between *Python Programming* and *Operating System*, 0.26 between *Python Programming* and *Linear Algebra*, 0.14 between *Database Principle* and *English*, -0.05 between *Python Programming* and *English*, -0.32 between *Database Principle* and *Literature and History*.
>
> ### Response to the “Additional Feedback” section:
> > 1.The authors should pay attention to the order of figures and tables. We usually place them according to the order of they are referred in the text.
>
> Thanks for your suggestion. We have adjusted related figures and tables in our revised paper.
>
> > 2. The authors should refine figure 4 (1) by enlarging the text font size.
>
> Thanks for your suggestion. We have adjusted Figure 4(a) in our revised submission.
>
> > 3. The authors should give more motivation explanation for using the meta learning strategy.
>
> Meta-learning aims to boost performance in novel tasks by training grounded in similar tasks. This paradigm integrates a dual-level learning schema: an extended learning of a meta-level model spanning multiple tasks and a rapid adaptation of a base-level model specific to each task [1]. In the context of cross-course scenario, each course, when approached as a distinct task, operates with a base-level model within the meta-learning framework. By leveraging cross-course information and employing meta-learning across multiple courses using a meta-level model, meta-learning can notably enhance the method's performance. A prominent tactic in meta-learning is the parameter-generating based approach, which employs a meta-learner to anticipate network parameters [2]. For our specific task, it is essential to establish a unique mapping function between the source and target domains for every student. Consequently, parameter-generating based methods are fitting as the meta-learner in this context. In our methodology, meta-learning enables the derivation of unique mapping function parameters tailored to each student, aligning with the observation that students possess distinct knowledge transfer abilities in educational scenarios.
>
> **Reference**
>
> [1] Munkhdalai, Tsendsuren, and Hong Yu. "Meta networks." International conference on machine learning. PMLR, 2017.
>
> [2] Zhu, Yongchun, et al. "Personalized transfer of user preferences for cross-domain recommendation." Proceedings of the Fifteenth ACM International Conference on Web Search and Data Mining. 2022.

---

> > ### Comment · Reviewer_ToYw · 2023-08-22
> >
> > Thanks for the authors' responses. My concerns are all solved.

---

### Official Review · Reviewer_T6X2 · 2023-07-23
**Comments**

**Rating:** 4
**Confidence:** 4
**Correctness:** No correctness issues.
**Clarity:** The paper is easy to follow.

**Strengths:**

1. The paper is well motivated and easy to read and follow.
2. The authors provide detailed data descriptions on the proposed dataset and show a modeling framework on this dataset.
3. Several experiments are conducted and results are evaluated via different metrics.

**Additional Feedback:**

No

**Documentation:**

Well documented.

**Ethics:**

No ethical issue.

**Limitations:**

1. The authors claim that the proposed dataset is student-centered. From lines 48-49, the word "student-centered" means student response logs. Every KT or CD related datasets have student responses. What's the speciality in this dataset? Furthermore, what's the real difference between "diverse" and "immense"?
2. After checking the datasets, it seems all the datasets are CS related data, even for the non-programming datasets. I think the "dataset diversity" is over claimed.
3. Section CCLMF is very confusing. In previous sections, the paper introduce a datasets contains lots of features. All of sudden, it comes to CCLMF which seems to have no directly relations with previous section. It's not clear how the CCLMF utilizes the rich information provided in the proposed dataset. The NCD model used in CCLMF is a bit old and only relies on very standard ID features.
4. In Section 5.3, the authors reported several baselines of KT and CD. It seems these baselines don't consider the diverse and immense features in the proposed dataset. These results are disconnected with the features of PTADisc.

Minors:
1. Footnotes of PTA are duplicated at both page 2 and page 3.
2. The courses in Figure 4(a) are Python, DS, C++, Java and C while courses in Figure 4(b) are compDB, Linux and Probability. I am a bit confused about this. Why they are different?

**Opportunities For Improvement:**

See Limitations section.

**Relation To Prior Work:**

It's not clear the real difference compared to previous work.

**Summary And Contributions:**

In this paper, the authors propose a new educational datasets for both the CD and KT tasks. The datasets contains student programming records of different courses. A modeling framework is proposed to illustrate how to do ML tasks using this dataset.

---

> ### Author Response · Authors · 2023-08-15
> **Individual Response to Reviewer T6X2 (Part A)**
>
> We express our gratitude to reviewer T6X2 for their valuable feedback and recommendations, which have greatly contributed to the improvement of our manuscript. However, we would like to address some potential misunderstandings and hope to clarify the contribution of our proposed dataset and method.
> ### Response to the “Relation To Prior Work” section:
> > 1. It's not clear the real difference compared to previous work.
>
> We summarize the main difference of the work compared to previous work as follows:
> - Immense: In terms of data scale, PTADisc is the largest educational dataset, containing over 680 million student response logs and providing various courses with different data scales.
> - Cross-course: PTADisc's most prominent feature lies in its extensive inclusion of cross-course information, a characteristic unparalleled by any other dataset and it is the first dataset to support cross-course analysis. It is worth emphasizing that we have also proposed a novel approach based on this cross-course characteristic.
> - Diversity: PTADisc encompasses valuable data such as programming logs, problem sets, student groups, and non-binary scores, which are not commonly found in other datasets.  For detailed information about data types, please refer to General Response 2. PTADisc can bring further research directions in the field of education. For detailed applications of PTADisc, please refer to General Response 3.
>
> ### Response to the “Opportunities For Improvement” section:
> > 1. The authors claim that the proposed dataset is student-centered. Every KT or CD related datasets have student responses. What's the speciality in this dataset?Furthermore, what's the real difference between "diverse" and "immense"?
>
> The term 'student-centered' primarily describes a characteristic of the dataset construction process. It emphasizes that our dataset is built and organized around students' responses, ensuring consistency in all information related to student activities. It is in contrast with 'knowledge-centered' datasets like MOOCCubeX which constructs datasets around knowledge concepts. Although MOOCCubeX also includes students' responses and can be used for CD and KT tasks, the quantity and quality of responses cannot be guaranteed.
>
> We have attempted to utilize MOOCCubeX for CD and KT tasks, but encountered two issues: 1. It required additional data preprocessing procedures to obtain a task-specific dataset; 2. The students' response logs on MOOCCubeX were found to be incomplete. Most CD and KT methods rely on the problem-knowledge concept relationship matrix (Q-matrix). But in MOOCCubeX, the knowledge concept annotations for problems are incomplete due to the keyword matching-based annotation process. As a result, the incomplete annotations limit the ability to utilize the full student response logs for a comprehensive diagnosis of their learning progress. In contrast, the student-centered PTADisc provides complete records of students' responses, making it highly suitable for KT and CD tasks.
>
> The term 'Immense' highlights the significant volume of data contained in our dataset.  While the term 'diversity' emphasizes the rich variety of data types, such as programming logs, problem sets, student groups, and non-binary scores, enabling the dataset to be applicable to a wide range of applications. Moreover, it is essential to highlight that the most significant feature of PTADisc is its unique 'cross-course' characteristics, which are not found in any previous educational datasets.
>
>
> > 2. After checking the datasets, it seems all the datasets are CS related data, even for the non-programming datasets. I think the "dataset diversity" is over claimed.
>
> As mentioned above, the term 'diversity' emphasizes the wide range of data types rather than the categories of courses. This diversity enhances the dataset's applicability across a broad spectrum of applications.  For more detailed information about data types and potential applications, please refer to General Response 2 and General Response 3.
>
> Our dataset encompasses students' learning records in various courses. These courses include subjects from the field of computer science, such as *Python Programming* and *C++ Programming*. Additionally, we have also included courses from the field of mathematics, such as *Discrete Mathematics* (with $1,694$ students and $125,378$ response logs) and *Probability and Statistics* (with $557$ students and $46,106$ response logs). Moreover, we have recently incorporated six courses related to social sciences and humanities, including *English* (with $1,952$ students and $139,328$ response logs) and *Literature and History* (with $2,306$ students and $126,748$ response logs), during the rebuttal period. For more detailed information about the course categories, please refer to General Response 1 and Table 1 in the Supplementary Material.

---

> > ### Author Response · Authors · 2023-08-15
> > **Individual Response to Reviewer T6X2 (Part B)**
> >
> > > 3. Section CCLMF is very confusing. In previous sections, the paper introduce a datasets contains lots of features. All of sudden, it comes to CCLMF which seems to have no directly relations with previous section. It's not clear how the CCLMF utilizes the rich information provided in the proposed dataset. The NCD model used in CCLMF is a bit old and only relies on very standard ID features.
> >
> > The most notable attribute of PTADisc is its provision of abundant cross-course information, which plays a crucial role in analyzing patterns of students' learning behaviors across different courses. Leveraging this attribute, we proposed CCLMF, a method that utilizes the relationships between students' proficiency levels across courses to alleviate the difficulty of diagnosing student knowledge states in cold-start scenarios. Thanks for the suggestion. To enhance the coherence of the text, we have included additional explanations at the conclusion of Section 3.5 and the beginning of Section 4.
> >
> > Next, we will explain why we choose NCD as the base model of CCLMF framework. Firstly, we would like to point out that research in the field of CD is not very active. Within this context, NCD is simple but the most representative work in CD. The limited research in this area is primarily attributed to the lack of availability of educational-related datasets that contain relevant information. It is worth mentioning that there are some related datasets, such as the one used in EERNN [1], that are not open-sourced. The diversity feature of PTADisc addresses this limitation to some extent. By providing a dataset that compensates for this scarcity, PTADisc opens up new possibilities and opportunities in the domain. For more detailed applications of PTADisc, please refer to General Response 3.
> >
> > > 4. In Section 5.3, the authors reported several baselines of KT and CD. It seems these baselines don't consider the diverse and immense features in the proposed dataset. These results are disconnected with the features of PTADisc.
> >
> > The main purpose of the baseline experiments in Section 5 is to demonstrate our dataset's reliability and generalizability across various existing methods, which is not associated with the 'diverse' feature.
> >
> > For the 'diverse' feature: due to the limited information in the previous datasets, the data features used by these existing methods were also relatively simple. Because we only applied our dataset to existing baseline methods, the 'diverse' feature could not be demonstrated. As for detailed future research directions of PTADisc's diverse feature, please refer to General Response 3.
> >
> > For the 'immense' feature: we have selected four courses with varying data scales, which aligns with the 'immense' feature mentioned in Section 3.4 that this range of data scales presents researchers with a multitude of options for personalized learning studies.
> >
> > > 5. Footnotes of PTA are duplicated at both page 2 and page 3.
> >
> > Thanks for pointing out. We have removed the footnotes on page 3.
> >
> > > 6. The courses in Figure 4(a) are Python, DS, C++, Java and C while courses in Figure 4(b) are compDB, Linux and Probability. I am a bit confused about this. Why they are different?
> >
> > The difference between the courses in these two figures is because they are designed for different purposes.
> > In Figure 4(a), we aim to illustrate the interrelationships among several courses. To analyze the characteristics of cross-course more effectively, we opted for more representative datasets, thus selecting several datasets with a higher number of overlapping students and extensive interaction records.
> > On the other hand, Figure 4(b) demonstrates the overall statistics of PTADisc and highlights the datasets used for the CD and KT tasks in Section 5. In Section 5, we intend to analyze the performance of CD and KT in datasets of varying scales. Hence, we have selected courses spanning a range of scales in Figure 4(b).
> >
> >
> > **Reference**
> >
> > [1] Su, Yu, et al. "Exercise-enhanced sequential modeling for student performance prediction." Proceedings of the AAAI Conference on Artificial Intelligence. Vol. 32. No. 1. 2018.

---

> > > ### Comment · Reviewer_T6X2 · 2023-08-16
> > >
> > > Thank you for your detailed response. I have read your replies and your paper one more time.
> > >
> > > However, I am not quite convinced. You main claim is the PTADisc is not only large but "rich". Rich means it contains lots of information/relations besides student responses. I cannot see how you use these "rich information". From my understanding, all your experiments are based on ids (student id, question id, knowledge concept ids) and student responses.
> > >
> > > In terms of your claim that "The limited research in this area is primarily attributed to the lack of availability of educational-related datasets that contain relevant information", this is not quite true. The difficulty part is how to utilize all these rich information.

---

> > > > ### Author Response · Authors · 2023-08-18
> > > > **Reply to Reviewer T6X2**
> > > >
> > > > Thank you for your prompt feedback. We appreciate your insights. However, we would like to clarify the main contribution of PTADisc, which lies in its **'cross-course' feature** rather than 'rich information'. PTADisc is the first dataset that supports cross-course analysis, as we have constructed a **cross-course subdataset** with $29,454$ students enrolled in five courses. This cross-course feature of PTADisc has been utilized by the proposed CCLMF, which analyzes the performance of students with learning records in multiple courses. This simple and intuitive method brings significant improvement in cold-start scenarios.
> > > >
> > > > Regarding the abundance of information in PTADisc, we are actively exploring strategies to effectively leverage this rich dataset. However, given the diverse nature of the information, it is not feasible to fully utilize all the available data within the scope of a single study. Therefore, we are making careful considerations and selecting the most impactful information, specifically the cross-course nature of PTADisc, for our analysis in this work.
> > > >
> > > > While existing educational datasets may contain some information, they generally lack the comprehensive nature of PTADisc. For a detailed comparison, please refer to Table 5 in our paper. From the table, we can observe that the most unique feature of PTADisc is its cross-course information, which is valuable for analyzing student performance across multiple courses.

---

> > > > ### Author Response · Authors · 2023-08-23
> > > >
> > > > We thank the reviewer for your helpful suggestions and insightful comments. Please feel invited to engage with us if you have additional problems. We hope our responses and revisions have adequately addressed your concerns, and we would like to politely ask the reviewer to re-evaluate our work. Thanks again.

---

### Author Response · Authors · 2023-08-15
**General Response (Part A)**

We express our gratitude to the reviewers for their constructive comments and suggestions.  In light of the feedback, we have revised the manuscript accordingly. General queries are addressed in the sections below, while specific queries from each reviewer are addressed in individual replies.

### 1. The course categories included in PTADisc
**The central contribution of our dataset is the cross-course information of students with learning records in mutiple courses.** And PTADisc is the first dataset to support **cross-course analysis**.

Because PTA is a programming teaching assistant platform, the majority of courses in PTADisc are computer science related. In addition to that, PTADisc also includes courses related to mathematics (e.g., *Discrete Mathematics* with $1,694$ students and $125,378$ response logs, and *Probability and Statistic* with $557$ students and $46,106$ response logs) and some courses related to social sciences and humanities (e.g., *English* with $1,952$ students and $139, 328$ response logs, and *Literature and History* with $2,306$ students and $126,748$ response logs). Specifically, during the rebuttal period, we have made an effort to enhance the course offerings on PTADisc by adding six additional courses, including *English*, *Literature* and *History*, *Psychology*, *Japanese*, *Politics*, and *Tourism*. For more detailed information on the course categories, please refer to Table 1 in our Supplementary Material.

It is important to note that while PTADisc encompasses a wide range of course categories, our primary focus and emphasis in this work is on the comprehensive selection of computer science courses.


### 2. Detailed data type and format
The raw data obtained from the PTA platform underwent an initial preprocessing step to transform it into intermediate **structured data**. Subsequently, the data was reorganized based on the specific task formats to create **task-specific data**. All data is organized by course.

In the following, we will provide an explanation of the data formats to enhance understanding. Additionally, we have included examples of course, problem, and response logs in Tables 1, 2, and 3 of our revised paper. These examples are intended to provide concrete illustrations of the data structure. For more detailed information of the data format, we provide a comprehensive description on our [github page](https://github.com/wahr0411/PTADisc).

#### (1) Structured Data Format
The structured data for each course contains three parts.

- Non-programming Data

Including problem_info.csv and response_logs.csv, where each entry of the former is presented as (problem_set_problem_id, problem_id, knowledge_id, full_score) and each entry of the latter is presented as (submission_id, student_id, create_at, problem_type, score, problem_set_id, problem_set_problem_id, status).

- Programming Data

Including a response logs file, where each entry consists of the following fields: (submission_id, student_id, create_at, language, score, problem_set_problem_id, problem_id, skill_id, code, response, time_consume, memory_consume).

- Other Information

Additional information of student group and problem sets are also provided in .csv files, recording the student group members (student_id, student_group_id), opened problem sets for student groups (student_group_id, problem_set_id), problem set's start time and end time (problem_set_id, start_time, end_time).

Please note that some courses include both non-programming and programming data, while others only include one of them. For more details, please refer to Table 1 in Supplementary Material.

We also provide global knowledge concepts which are stored in a .csv file and organized in a tree-like structure, with each row containing (knowledge_id, knowledge_name, parent_knowledge_id). Additionally, we provide various mapping files in .json format, which store the correspondences between problem_ids and the corresponding attributes such as difficulty, knowledge concept, reference count, and problem type. We also include mappings between psp_id and full score or problem_id.

#### (2) Task-specific Datasets
The task-specific datasets only consider non-programming data.
- For the CD task, each course includes two main files: a problem-knowledge relation file, where each entry is represented as (problem_id, knowledge_ids) pairs, and a students' response log file, where each entry is represented as (student_id, item_id, score). An 'info.json' file is provided for each course, containing information on the number of students, problems, knowledge concepts, and the number of response logs in that course. Furthermore, we provide training, validation, and test splits for our baseline experiments.
- For the KT task, each course includes a response log file, where each line consists of the following fields: (submission_id, student_id, create_at, score, problem_id, reference_count, skill_id, difficulty).

---

> ### Author Response · Authors · 2023-08-15
> **General Response (Part B)**
>
> ### 3. The applications of PTADisc
> We have supplemented the potential applications of PTADisc in Section 3.5 in our revised paper. PTADisc not only supports existing CD and KT methods but also supports:
>
> (1) Prerequisite discovery [1], which refers to the task of identifying and establishing the sequence or order in which concepts or topics should be learned or presented, ensuring that foundational concepts are understood before more advanced ones.
>
> (2) Computerized adaptive testing [2], which is an emerging testing format in many standardized examinations, aims to rapidly and accurately diagnose a candidate's level of knowledge mastery through personalized test items/exercises.
>
> (3) Educational recommendation [3], which is capable of making appropriate learning suggestions to students,  considering the interactions between students and questions.
>
> (4) Cross-course research. PTADisc provides cross-course information, enabling the study of how students perform in different classes. We have conducted a cross-course study addressing the cold-start issue in Section 4.
>
> With various information provided, PTADisc also has the potential to support the following research directions and opportunities:
>
> (1) It provides non-binary student performance data. Most existing CD and KT methods tackle this as binary classification (wrong/right answer). Non-binary grades allow for more regression-focused investigations.
>
> (2) It provides information on different problem types, aiding research of CD and KT.
>
> (3) It provides data on problem difficulty, enabling to study how the level of difficulty of exercises relates to students' final learning outcomes.
>
> (4) It provides problem set specifics, like submission time, enabling modeling of students' learning habits based on when they submit their work.
>
> (5) It provides information on student groups, facilitating the assessment of teaching quality within classes and group-level educational analysis.
>
> (6) It provides information on programming exercises, including detailed code submissions and records, enabling in-depth research into programming-related studies.
>
> **Reference**
>
> [1] Pan, Liangming, et al. "Prerequisite relation learning for concepts in moocs." Proceedings of the 55th Annual Meeting of the Association for Computational Linguistics (Volume 1: Long Papers). 2017.
>
> [2] Bi, Haoyang, et al. "Quality meets diversity: A model-agnostic framework for computerized adaptive testing." 2020 IEEE International Conference on Data Mining (ICDM). IEEE, 2020.
>
> [3] Huang, Zhenya, et al. "Exploring multi-objective exercise recommendations in online education systems." Proceedings of the 28th ACM International Conference on Information and Knowledge Management. 2019.
>
>
>
> ### Summary of revised submission
> All changes in the revised paper are in blue. The new additions to the paper are listed below.
>
> 1. We added six courses including English, Literature and history, Psychology, Janpanese, Politics, and Tourism. And we modified the text description related to statistical data, updated Figure 4(b) of the paper and Figure 1 and Table 1 in the Supplementary Material.
>
> 2. We supplemented the description of our sourse platform PTA at the begining of Section 3.
>
> 3. We provided data examples of structured data in Table 1, 2 and 3. And we added related description in Section 3.2.2.
>
> 4. We added a subsection 3.5 to discuss the applications and possible research directions supported by PTADisc. And we added definitions of these applications in Supplementary Material A.3.
>
> 5. We added additional explanations at the beginning of Section 4 to enhance the coherence of the text.
>
> 6. We supplemented the reason for choosing Python Programming and Java Programming for our cross-course study in Section 4.3.
>
> 7. We conducted additional experiments of CCLMF between the selected five courses in Figure 4(a). The results are added in the Supplementary Material E.

---

### Decision · Program_Chairs · 2023-09-22

**Decision:**

Accept (Poster)

**Comment:**

Reviews are positive, although some reviewers struggled with fully understanding the paper. This seems to be a function of how the paper is written and presented. Reviewers have provided help on how to better explain to readers what the work is doing. Notably, the authors' main point that the dataset is focused on supporting "cross-course analysis" – and does not offer rich, diverse content outside of this paradigm – should be made more clear in the paper.  After discussions and clarifications, the reviews support the following ideas:

Strengths:
- Paper provides a new large, unique dataset.
- The dataset provides a novel type of information: detailed cross-course information.
- Dataset can be valuable to several communities (data mining, education).
- Dataset can be useful for several tasks.
- Authors demonstrate its utility for proficiency prediction well.

Opportunities For Improvement:
- Outline potential applications of the dataset across various tasks.
- Provide some analysis and examples on students’ behavior patterns across difference courses.
- Address issues with label definitions and their implications.

Limitations:
- Addressed well by authors.

Correctness:
- Reasonable.

Clarity:
- Reasonable, although there are suggestions throughout on what to improve in a final version.

Relation To Prior Work:
- Done well.

Documentation:
- Done well.